# CONTINUAL LEARNING IN THE PRESENCE OF SPURIOUS CORRELATIONS: ANALYSES AND A SIMPLE BASELINE

**Donggyu Lee**[1*]**, Sangwon Jung**[2*] **& Taesup Moon**[2,3†]
[1]Department of Electrical and Computer Engineering, Sungkyunkwan University
[2]Department of Electrical and Computer Engineering, Seoul National University
[3]ASRI / INMC / IPAI / AIIS, Seoul National University
`ldk308@skku.edu`, `{s.jung,tsmoon}@snu.ac.kr`

## ABSTRACT

Most continual learning (CL) algorithms have focused on tackling the stability-plasticity dilemma, that is, the challenge of preventing the forgetting of past tasks while learning new ones. However, we argue that they have overlooked the impact of knowledge transfer when the training dataset of a certain task is *biased* — namely, when the dataset contains some *spurious correlations* that can overly influence the prediction rule of a model. In that case, how would the dataset bias of a certain task affect the prediction rules of a CL model for future or past tasks? In this work, we carefully design systematic experiments using three benchmark datasets to answer the question from our empirical findings. Specifically, we first show through two-task CL experiments that standard CL methods, which are oblivious of the dataset bias, can transfer bias from one task to another, both forward and backward. Moreover, we find out this transfer is exacerbated depending on whether the CL methods focus on stability or plasticity. We then present that the bias is also transferred and even accumulates in longer task sequences. Finally, we offer a standardized experimental setup and a simple, yet strong plug-in baseline method, dubbed as group-class **B**alanced **G**reedy **S**ampling (BGS), which are utilized for the development of more advanced bias-aware CL methods[1].

## 1 INTRODUCTION

Continual learning (CL) is essential for a system that needs to learn (potentially increasing number of) tasks from sequentially arriving data. The main challenge of CL is to overcome the *stability-plasticity* dilemma (Mermillod et al., 2013); when a CL model focuses too much on stability for remembering past tasks, it would suffer from low plasticity for learning a new task (and vice versa). Recent deep neural networks (DNNs) based CL methods (Kirkpatrick et al., 2017; Jung et al., 2020; Li & Hoiem, 2017) attempted to address the dilemma by devising mechanisms to attain stability while improving plasticity thanks to the *knowledge transferability* (Tan et al., 2018), which is one of the standout properties of DNNs. Namely, while maintaining the learned knowledge, the performance on a new task (resp. past tasks) is improved by transferring knowledge of past tasks (resp. a new task). Such phenomena are called forward and backward transfer, respectively.

However, most of such DNN-based CL approaches have not explicitly considered a more realistic and challenging setting in which the *dataset bias* (Torralba & Efros, 2011) exists; *i.e.*, a training dataset contains unintended correlations between some spurious features and class labels. In such a case, it is widely known that DNNs often dramatically fail to generalize to the test data without the correlation due to overly relying on the spurious features (Geirhos et al., 2020). Since the failure cases by such a so-called shortcut learning are being commonly observed in various application areas (Geirhos et al., 2020), it is also necessary to thoroughly study the effect of spurious correlations in the CL setting.

---

[*]Equal contribution
[†]Corresponding author
[1]Code is available at `https://github.com/DQle38/BGS`.

To that end, we claim that the effect of spurious correlations becomes significant in CL due to the *bias transfer* issue. In a recent study (Salman et al., 2022), it is shown that during transfer learning, the bias learned by a pre-trained model is (forward) transferred to the downstream model even when it is fine-tuned with unbiased downstream task data. In CL, this issue can be potentially *exacerbated* since typical CL methods, which are oblivious of the bias transfer, would prevent the forgetting of learned tasks that had the dataset bias. Moreover, the bias transfer can occur in *both* forward and backward directions as CL involves learning a sequence of tasks. For instance, consider an example of domain-incremental learning (Domain-IL) shown in Figure 1, in which the goal is to incrementally learn the classification model that predicts whether the face in the image is *young* or *old*. Now, we assume that only the training dataset for $T_t$ possesses the *gender* bias, that is, the spurious correlations exist between *young* and *female* as well as *old* and *male*. We then argue that when a naive CL method is used to update the model $h_t$ from $T_t$, the gender bias picked up by the model can encourage to

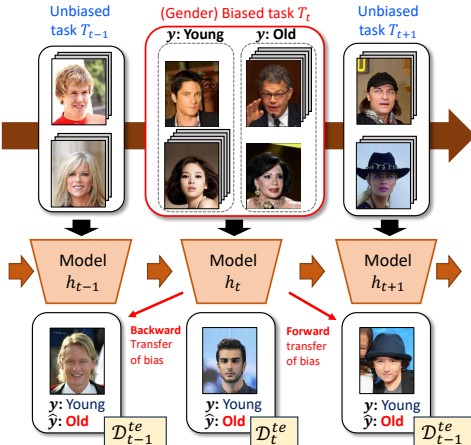

Figure 1: **Bias transfer in CL.** Three tasks sequentially arrive, in which each task has different distributions such as face images having blond hair, black hair, and hat. The CL models, $h_t$ and $h_{t+1}$, may falsely classify young man images in $T_{t-1}$ and $T_{t+1}$ to "Old" due to the bias transfer from $T_t$.

make gender-dependant predictions in the previous or future tasks (*e.g.*, $T_{t-1}$ or $T_{t+1}$) even if those tasks do not contain any dataset bias.

One might reason that the risk of such bias transfer can be expected since spurious features may be transferred to other tasks in a similar fashion as regular features. Nonetheless, to the best of our knowledge, an extensive investigation on this issue has not been carried out in the CL setting. Therefore, in this paper, we conduct thorough and comprehensive experiments to demonstrate the substantial quantitative evidence that the bias transfers in CL, both forward and backward, indeed exist and significantly affect model predictions. Specifically, we design three representative CL experiment settings with several benchmark datasets containing spurious correlations and employ well-defined metrics for CL and bias, as well as various CL baselines (Section 3). We then carry out simple two-task CL experiments and identify that when a typical CL method focuses on the stability (resp. plasticity), the forward (resp. backward) transfer of bias occurs (Section 4). Furthermore, for a longer sequence of tasks, we show that the bias is also transferred to much earlier or later tasks and even *accumulates* when naively applying CL methods (Section 5). Finally, we provide a standard experimental setting including benchmark bias-aware CL scenarios and evaluation protocols, along with a simple yet strong plug-in baseline method, dubbed group-class **B**alanced **G**reedy **S**ampling (BGS), which can aid in developing a more advanced bias-aware CL method.

## 2 RELATED WORK

**Continual learning (CL).** Assuming that tasks are well-separated, CL scenarios are typically divided into three categories (Van de Ven & Tolias, 2019): task-incremental learning (Task-IL), domain-incremental learning (Domain-IL), and class-incremental learning (Class-IL). Both the Task-IL and Class-IL assume each task has a disjoint set of labels, and the task identity is provided during training, but they differ in whether the identity is available (Task-IL) or not (Class-IL) at test time. In Domain-IL, the class set of tasks always remains the same, but only input distributions vary as the number of tasks increases.

Recent CL methods can also be classified into three groups based on how they prevent forgetting of the previously learned tasks (De Lange et al., 2021): regularization-based, rehearsal-based, and parameter isolation-based methods. Regularization-based methods add regularization terms for penalizing deviation from past models (Kirkpatrick et al., 2017; Li & Hoiem, 2017; Jung et al., 2020). Rehearsal-based methods store some data points from past tasks in a small exemplar memory and replay them while learning the current task (Chaudhry et al., 2019; Lopez-Paz & Ranzato, 2017). Parameter isolation-based methods (Mallya & Lazebnik, 2018; Ye & Bors, 2021) allocate model

parameters separately for each task by masking out previous task parameters and updating only the remaining parameters for learning a new task.

**Spurious correlations and debiased learning**. The problem of addressing spurious correlations in machine learning models has recently become a popular research topic (Geirhos et al., 2020). Enormous studies have identified the issue of spurious correlations in various domains (Geirhos et al., 2019; Garrido-Muñoz et al., 2021) and strived to find out the source of the correlations Scimeca et al. (2021). In addition, numerous techniques to solve these issues have been developed by utilizing additional annotations for spurious correlation (Sagawa et al., 2020) or prior knowledge for the learning mechanism of spurious features (Liu et al., 2021; Kirichenko et al., 2023). Nonetheless, all the works above have not considered the existence of spurious correlations in CL.

**CL considering spurious correlations**. While most state-of-the-art CL methods have not considered the effect of spurious correlations, there are two notable exceptions (Lesort, 2022; Jeon et al., 2023). Lesort (2022) considers the issue of local spurious features (LSF) which correlates with class labels in a particular task but does not correlate when data samples from all tasks are considered. Jeon et al. (2023) find out that forgetting the biased knowledge obtained from previous tasks is conducive to LSF problems, thereby devising a method that encourages such forgetting while keeping other information maintained. However, both of the works only focus on Domain-IL and lack in-depth analyses for more general bias transfer problems in CL.

## 3 EXPERIMENTAL SETUP

### 3.1 PROBLEM SETTING AND DEFINITION

**Notation**. We consider the *bias-aware* CL setting, which is composed of a sequence of classification tasks in which dataset biases may exist. Each data sample in the $t$-th task, $T_t$, consists of the tuple $(x_i, a_i, y_i)$ where $a_i \in \mathcal{A}$ and $y_i \in \mathcal{Y}_t$ are the *group* and *class* label of an input $x_i$, respectively. The group label is defined by an attribute of $x_i$ that may be spuriously correlated with the class label, *i.e.*, cause *dataset bias*. Unless otherwise noted, we consider a single type of dataset bias for the simplicity of analysis. In addition, the sets of class labels, $\{\mathcal{Y}_t\}$, can be identical or not across $t$, depending on the CL scenarios.

**Metrics for model bias**. We measure the extent of *bias* of a classification model with respec to a group label by computing the number of prediction flips given counterfactual inputs with a flipped group label given correctly classified inputs. Formally, *Bias-flipped Mis-classification Rate* (BMR) is defined as a metric for the bias of a model $h$:

$$\mathrm{BMR}(h) = \frac{\sum_{\{x_i \in \mathcal{D}|h(x_i)=y_i\}} \mathbb{I}(h(x_i^*) \neq y_i)}{|\{x_i \in \mathcal{D}|h(x_i) = y_i\}|}, \tag{1}$$

in which $x_i^*$ is the counterfactual input with a flipped group label. While BMR is an ideal bias metric, it can be challenging to generate counterfactual samples in some real-world datasets like facial images. In such a case, we employ the *Difference of Classwise Accuracy* (DCA) (Berk et al., 2021) as a surrogate metric for BMR. DCA[2] indicates the gap in accuracy among different groups, which can be roughly considered as an approximation of BMR at the group level. Note that high BMR and DCA indicate a high degree of existence of spurious correlations/bias of a model.

**Metric for the relative focus on the Stability-Plasticity**. Since there exists the stability-plasticity trade-off, we also employ *Normalized $\mathcal{F} - \mathcal{I}$* as a metric to gauge the relative emphasis on stability and plasticity of a CL method. For a concrete definition, let $h_t$ and $h_t^*$ be the models learned up to $T_t$ by a CL baseline and the naive fine-tuning, respectively. Then, we introduce the forgetting and intransigence measures, $\mathcal{F}_t$ and $\mathcal{I}_t$, (Cha et al., 2021) which are commonly used for quantifying the stability and plasticity of $h_t$, respectively, as follows:

$$\mathcal{F}_t := \frac{1}{t-1} \sum_{j=1}^{t-1} \left[ \max_{l \in [j, t-1]} \mathrm{A}(h_l, \mathcal{D}_j) - \mathrm{A}(h_t, \mathcal{D}_j) \right], \quad \mathcal{I}_t := \frac{1}{t} \sum_{j=1}^{t} [\mathrm{A}(h_j^*, \mathcal{D}_j) - \mathrm{A}(h_j, \mathcal{D}_j)], \tag{2}$$

in which $\mathrm{A}(h, \mathcal{D}_t)$ denotes the accuracy of a model $h$ on the test set $\mathcal{D}_t$ for the task $T_t$. Then, with some pre-selected hyperparameters, we compute $\mathcal{F}_T - \mathcal{I}_T$ for each hyperparameter of each CL

---

[2]The formal definition of DCA is provided in Appendix C.

method after learning the final task $T_T$. Since CL methods have different degrees of $\mathcal{F}_T - \mathcal{I}_T$, we re-scale these values with the min-max normalization for each CL method to obtain the "Normalized $\mathcal{F} - \mathcal{I}$" metric. From the definition, we can observe that the small (resp. large) Normalized $\mathcal{F} - \mathcal{I}$ value indicates more focus on stability (resp. plasticity).

**How to investigate bias transfer**. As illustrated in Figure 1, when the dataset bias of a certain task adversely affects the other tasks in CL scenarios, we refer to this phenomenon as *bias transfer*. We can systematically assess the presence and magnitude of bias transfer with the metrics defined above. Concretely, assume a CL model $h$ learns a sequence of tasks via a CL method. Next, we intentionally intensify the dataset bias of a certain task, $T_i$, and learn the modified sequence of tasks to obtain another model $h'$. Then, the degree of forward and backward bias transfers can be measured by comparing the biases of $h$ and $h'$ on the tasks other than $T_i$. We note that the changes in the bias of the model should be evaluated under fixed Normalized $\mathcal{F} - \mathcal{I}$ metric of a CL method since the knowledge transfer would vary depending on the relative focus on the stability and plasticity.

## 3.2 Benchmark datasets

We use three benchmark datasets, Split CIFAR-100S, CelebA[2] (or CelebA[8]), and Split ImageNet-100C, which are applied for Task-IL, Domain-IL, and Class-IL, respectively. The following is the description of our datasets (for more details, see Appendix B.4.).

**Split CIFAR-100S** is a modification of Split CIFAR-100 (Chaudhry et al., 2019), which randomly divides CIFAR-100 (Krizhevsky, 2009) into 10 tasks with 10 distinct classes. As in Wang et al. (2020), we introduce the "color" bias into each task by disproportionately converting some color images to grayscale. Specifically, within each task, half of the classes are skewed towards the grayscale group and the other half towards the color group, with a given skew-ratio $\alpha \geq 0.5$. We set seven levels of bias (0-6) by dividing the range of skew-ratio from 0.5 to 0.99 evenly on a log scale for systematic control of the degree of bias.

**CelebA** (Liu et al., 2015) contains more than 200K face images, each annotated with 40 binary attributes. It is notorious for containing representation biases towards specific attributes, such as race, age, or gender (Torfason et al., 2017; Fabbrizzi et al., 2022). Unless otherwise specified, we use "male" and "young" attributes as the group and class labels, respectively (please refer to Table B.2). We additionally select one or three other attributes to divide CelebA into two or eight tasks (which are used in Section 4 and Section 5-6, respectively), denoted as CelebA[2] and CelebA[8], respectively. We again set seven levels of the dataset bias as in Split CIFAR-100S[3].

**Split ImageNet-100C** is made from Split ImageNet-100 that consists of 10 tasks with disjoint 10 classes, randomly sampled from ImageNet-1K (Russakovsky et al., 2015). We disproportionately injected Gaussian noise into $(\alpha \times 100)\%$ of the samples for one randomly selected class and only 1% of samples for other classes, given skew-ratio $\alpha \geq 0.01$. We similarly set seven levels of bias depending on the skew-ratio.

## 3.3 Continual learning and debiasing baselines

On the above datasets, we first evaluate two naive baselines, *freezing* that freezes the model learned from the previous task and *fine-tuning* that simply fine-tunes the model with new task data. Note they are the two extreme schemes that only focus on stability and plasticity, respectively. In addition, we compare six representative CL methods: LWF (Li & Hoiem, 2017) and EWC (Kirkpatrick et al., 2017) for regularization-based methods, ER (Chaudhry et al., 2019), iCaRL (Rebuffi et al., 2017), and EEIL (Castro et al., 2018) for rehearsal-based methods, and PackNet (Mallya & Lazebnik, 2018) for a parameter isolation-based method. We note that each CL method can control the stability-plasticity trade-off by adjusting its own hyperparameters such as the regularization strength, the size of the exemplar memory, or the pruning ratio. We further note that PackNet can be applied only for Task-IL and, iCaRL and EEIL for Class-IL. Additionally, we employ a widely used debiasing technique, Group DRO (Sagawa et al., 2020), as a baseline to naively combine with CL techniques to reduce the dataset bias. More implementation details are provided in Appendix B.

---

[3]With this setting, we have not considered the issue of LSF in Domain-IL (Lesort, 2022; Jeon et al., 2023), *i.e.*, for the case that the skew-ratio ranges from 0 to 0.5. For further discussion, please refer to Appendix D.

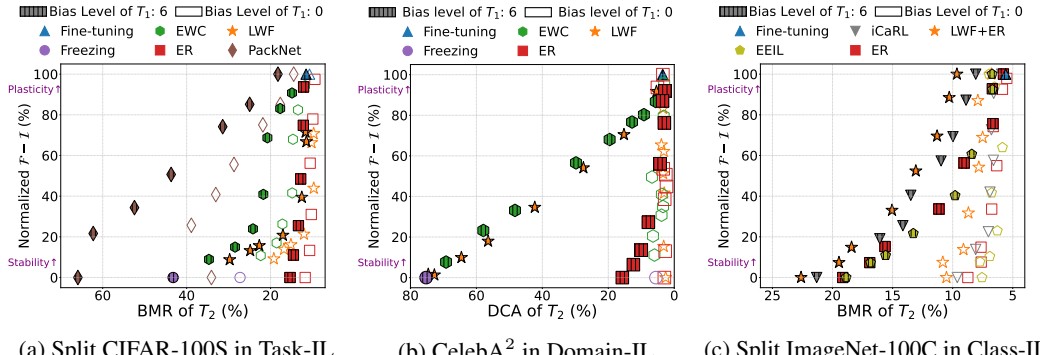

Figure 2: Forward transfer of bias in two tasks-continual learning.

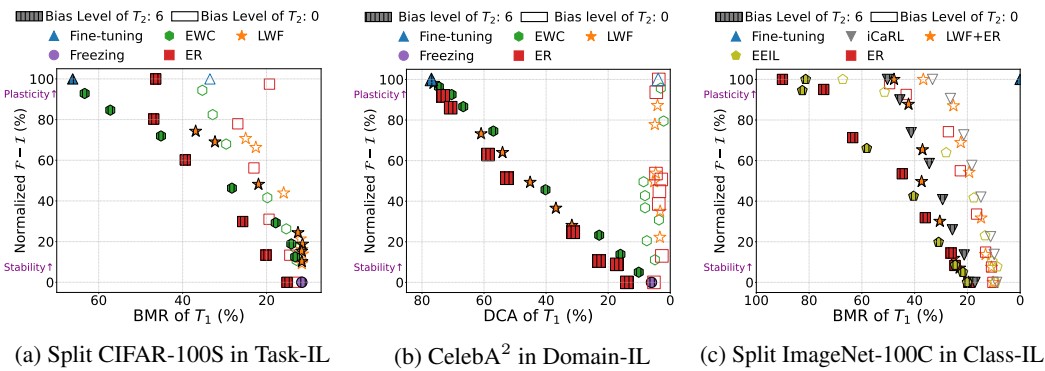

Figure 3: Backward transfer of bias in two tasks-continual learning.

## 4 CASE FOR CL WITH TWO TASKS

We begin our analysis by examining a simple setting: continual learning of two tasks. We aim to identify forward and backward transfers of bias for a CL model through quantitative analyses using our metrics. We also aim to understand how the CL methods aggravate each of these transfers.

### 4.1 FORWARD TRANSFER OF THE BIAS

To investigate the forward transfer of bias, we evaluated CL methods for two different bias levels for $T_1$ (*i.e.*, level 0 and 6), while that of $T_2$ is fixed to level $0^4$, *i.e.*, to be unbiased. Figure 2 reports the bias metric values for $T_2$ along with Normalized $\mathcal{F} - \mathcal{I}$ on the three datasets, each with a different CL scenario, after learning $T_2$. In the figure, we plot the results of each CL method by varying their own hyperparameters for controlling the stability-plasticity trade-off. Namely, the upper points on each plot represent a lower regularization strength, a smaller memory size, or a lower pruning ratio for each method, hence putting more emphasis on plasticity, and vice versa.

From the figures, we make the following observations. First, from the BMR gap of the two blue triangles in Split CIFAR-100S results (11.53% vs 10.72%), we observe that even with simple fine-tuning, the bias of $T_1$ adversely affects that of $T_2$, *i.e.*, forward transfer of bias exists, which is consistent with Salman et al. (2022). Second, for all CL methods, the gaps between colored and uncolored points for similar Normalized $\mathcal{F} - \mathcal{I}$ are larger than fine-tuning, and mostly increase as Normalized $\mathcal{F} - \mathcal{I}$ becomes lower. In other words, when $T_1$ is severely biased, the CL methods transfer such bias more than fine-tuning since they are designed to remember past tasks, and the level of transfer becomes severer as the CL methods more focus on stability. We additionally report the accuracy of $T_2$ in Appendix E.2 and observe similar accuracies between colored and uncolored points, especially for Split CIFAR-100S and Split ImageNet-100C, meaning that the bias gaps are not caused by the accuracy gaps. Finally, we notice that the bias transfer occurs even when the dataset

---

[4] We additionally report the results on Split CIFAR-100S when the bias level of $T_2$ is 2 or 4 in Appendix E.1 and observe similar trends from Figure 2 and 3.

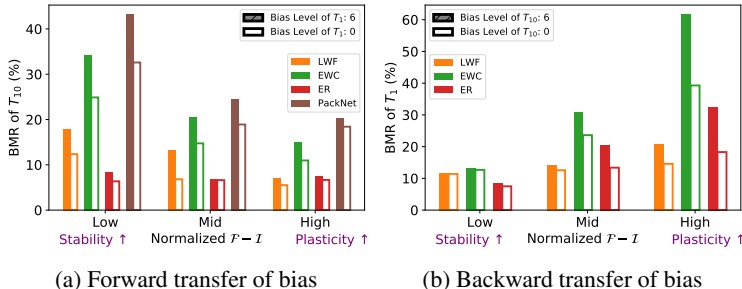

(a) Forward transfer of bias        (b) Backward transfer of bias

Figure 5: **Bias transfers in a sequence of 10 tasks on Split-CIFAR100S**. The BMRs of $T_{10}$ and $T_1$ are shown after learning up to $T_{10}$ by CL methods.

bias in $T_1$ is relatively mild, demonstrating that the bias transfer is not confined to cases where $T_1$ exhibits heavy bias. This finding is included in Appendix E.1.

## 4.2 BACKWARD TRANSFER OF THE BIAS

Now, we study the backward transfer of bias. Figure 3 compares the bias of a model for task $T_1$, of which the bias level is fixed to 0, after learning $T_2$. Similarly as in Section 4.1, the two different bias levels for $T_2$, *i.e.,* level 0 and 6, are tested. We omit the results for PackNet since PackNet does not change the predictions for $T_1$ after learning $T_2$ and hence, there is no backward transfer.

We clearly observe from Figure 3 that the trend is now opposite to Figure 2. Firstly, we observe that the bias gap for $T_1$ under similar Normalized $\mathcal{F} - \mathcal{I}$ is maximized by fine-tuning[5]. Also, for each CL method, the gap becomes severe as the Normalized $\mathcal{F} - \mathcal{I}$ increases. This means that the more CL methods prioritize plasticity over stability, the more bias obtained from $T_2$ is transferred to $T_1$, *i.e.*, the backward transfer of bias occurs more. We report the accuracy of $T_1$, and results for the milder bias of $T_1$ and $T_2$, in Appendix E.2 and E.1.

## 4.3 FEATURE REPRESENTATION ANALYSIS

In our pursuit to provide concrete evidence of the bias transfer, we analyze feature representations from the penultimate layer of a DNN-based model using the centered kernel alignment (CKA) with the linear kernel (Kornblith et al., 2019). CKA is an isotropic scaling-invariant metric, making it suitable for quantifying the similarity between two representations derived from a DNN model. Figure 4 compares the CKA values on Split CIFAR-100S for EWC under the two-tasks settings similar to Figure 2a. Namely, we evaluate models after learning $T_2$ by varying the regularization strength and the bias level of $T_1$. We then compute CKA between color and grayscale images in the test dataset of $T_2$. Therefore, a lower CKA value implies a greater dissimilarity in representations between two groups, signifying a higher color bias in the model. Figure 4 clearly shows that as the regularization strength increases and the bias of $T_1$

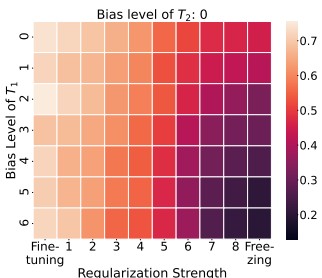

Figure 4: CKA values in $T_2$ on Split CIFAR-100S.

is more severe, CKA value decreases. Thus, the feature representation analysis again corroborates the forward transfer of the bias. For the case of backward transfer, please refer to Appendix E.3.

## 5 CASE FOR CL WITH A LONGER SEQUENCE OF TASKS

We now further investigate whether the bias transfers shown for two-task CL scenarios also generalize and are aggravated for a longer sequence of tasks. In order to enhance the clarity of the comparison,

---

[5]We observed that the models trained by fine-tuning in Class-IL classified all the samples in $T_1$ into one of the classes in $T_2$ and hence, it is infeasible to compute BMR since accuracy for $T_1$ is 0. Thus, we instead marked its BMR as 0% in the figure. In addition, we exclude the result of freezing in Class-IL since when learning the second task through freezing, only the linear classifier is learned for the second task while the classifier for the first task is fixed and hence, the learned model produces unexpected low stability and high plasticity.

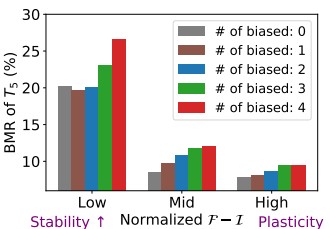 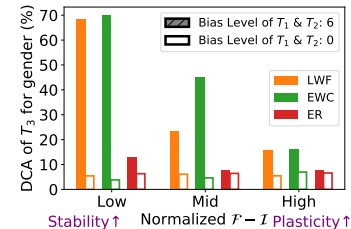 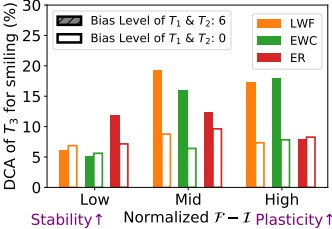

Figure 6: Accumulation of the same type of bias on Split CIFAR-100S.

Figure 7: Accumulation of the different types of bias on CelebA[8]. DCAs of $T_3$ for "gender" and "smiling" attributes are reported.

we simplify the visual format of all figures in this section. We divide a range of the Normalized $\mathcal{F} - \mathcal{I}$ into three equal intervals, and provide one result for each interval and CL method: the result for a hyperparameter that leads to the lowest Normalized $\mathcal{F} - \mathcal{I}$ within an interval.

## 5.1 BIAS TRANSFER IN LONGER SEQUENCES

Firstly, we consider a sequence of 10 tasks with similar settings as in Figure 2 and 3 on Split CIFAR-100S. In Figure 5a and 5b, we vary the bias level of $T_1$ and $T_{10}$ to 0 or 6, respectively, while the levels of all other tasks are fixed to 0. Correspondingly, the two plots show BMR of $T_{10}$ (resp. $T_1$) according to the bias of $T_1$ (resp. $T_{10}$) for each CL method. Thus, Figure 5a and 5b are for the forward and backward transfers, respectively.

The plots reveal an analogous trend with our two-task analyses. Namely, we observe that BMR of $T_{10}$ (resp. $T_1$) is mostly higher when the bias level of $T_1$ (resp. $T_{10}$) is 6, compared to when that of $T_1$ (resp. $T_{10}$) is 0. Furthermore, the gaps of BMR between colored and uncolored bars are at their widest when the Normalized $\mathcal{F} - \mathcal{I}$ is low (for Figure 5a) and high (for Figure 5b). Namely, it is obviously shown that both types of bias transfers also occur in longer sequences of tasks and can be aggravated by CL methods. We additionally display the accuracy of $T_{10}$ and $T_1$ for each plot in Appendix E.4 and notice that the accuracies are roughly the same, meaning that the gaps of BMR are due to the bias transfer, not due to accuracy gaps. Finally, we emphasize that the bias of a CL model indeed can *persist* even if the tasks in the middle are not biased, depending on the relative focus on stability and plasticity.

## 5.2 ACCUMULATION OF THE SAME TYPE OF DATASET BIAS

We now examine whether the *same* type of dataset bias for each task could get accumulated by CL methods. To this end, we take a sequence of 5 tasks with a bias level of 0 in Split CIFAR-100S. We then randomly select some of the four tasks except the last task and change the bias level of them to $4$[6] to make biased tasks. Figure 6 presents the BMR values according to the varying number of biased tasks when LWF is applied. From the results, it is evident that BMR of $T_5$ becomes higher as the number of biased tasks increases. This observation indicates that when tasks with the same type of dataset bias are continually added, their bias *accumulates* within the CL model, resulting in a greater negative impact on other tasks. Also, note that the bias accumulates more when CL focuses more on stability. Accuracy of $T_5$ and the results for backward transfer are provided in Appendix E.4 and E.5.

## 5.3 ACCUMULATION OF THE DIFFERENT TYPES OF DATASET BIAS

In order to investigate whether *different* types of dataset bias can also accumulate, we consider sequences of three tasks, each randomly selected from CelebA[8]. Specifically, in the training datasets of $T_1$ and $T_2$, we use "gender" and "smiling" attributes as group labels, respectively. Then, while fixing the bias levels of $T_3$ for both group labels to 0, we evaluate DCA of $T_3$ in terms of both group labels by varying the bias level of $T_1$ and $T_2$.

Figure 7 displays the degree of gender and smiling bias of models at $T_3$ after CL of three tasks with LWF, EWC, and ER. We observe that when $T_1$ and $T_2$ are biased towards gender and smiling,

---

[6]We used a lower level of bias (*i.e.*, 4), instead of 6, in order to more clearly display how BMR varies with the number of biased tasks.

respectively, DCA of $T_3$ gets mostly higher in terms of both group labels. Furthermore, we again notice from the left plot of Figure 7 that when CL methods more focus on stability, the forward transfer of bias becomes stronger. In contrast, in the right plot, the gap between colored and uncolored bars for LWF and EWC is larger when Normalized $\mathcal{F} - \mathcal{I}$ is in the middle regime, compared to the low regime. We infer that this might be because the dataset bias of $T_2$ is less likely to be picked up under lower plasticity when learning $T_2$, resulting in the bias being less transferred to $T_3$.

## 6 BIAS-AWARE CONTINUAL LEARNING

We demonstrated that the dataset bias of each task can be transferred by CL methods in various scenarios, strongly highlighting the need for a new CL approach that considers the dataset bias of each task. To that end, we provide a standardized experimental setup for developing a novel bias-aware CL method, including evaluation metrics, benchmark bias-aware CL scenarios, and a hyperparameter selection rule, together with a simple, yet effective plug-in baseline method, BGS. Based on these resources, we display the performance for typical CL methods and BGS in bias-aware CL scenarios, and pose discussions about future directions of bias-aware CL studies.

### 6.1 EXPERIMENTAL SETUP FOR DEVELOPING BIAS-AWARE CL METHOD

We construct three kinds of bias-aware CL scenarios for evaluation: 10-task sequences on Split CIFAR-100S and Split ImageNet-100C, and an 8-task sequence on CelebA[8]. The degree of bias for each task in Split CIFAR-100S and Split ImageNet-100C is controlled with a uniformly sampled skew-ratio by injecting the color or noise bias. On the other hand, we use CelebA[8] as it is without artificially adjusting the number of samples, in order to reflect a more realistic bias-aware CL scenario[7]. For evaluation metrics, we employ average accuracy and BMR (or DCA) across all tasks in each scenario. Using the bias-aware CL scenarios and metrics, we analyze the performance of naive CL methods and combinations between CL methods and the two debiasing techniques, Group DRO and BGS. We note that Group DRO combined with a CL method employs an integrated loss function of both methods. To consider practicality, all hyperparameters for each CL method and Group DRO are tuned based on the average accuracy and bias metric up to $T_3$, as presented in (Mai et al., 2022). We choose the best hyperparameters with the average accuracy for naive CL methods. For Group DRO combined with a CL method, we fix the best hyperparameters for applying the CL method and search for the best ones of Group DRO with the best bias metric.

### 6.2 GROUP-CLASS BALANCED GREEDY SAMPLING

BGS is inspired by two recently developed methods for CL and debiasing, GDumb (Greedy Sampler and Dumb Learner) (Prabhu et al., 2020) and DFR (Deep Feature Re-weighting) (Kirichenko et al., 2023). The main idea of both methods is to simply train a model using a small amount of balanced data. Specifically, GDumb utilizes a greedy sampler to collect balanced data over all classes in the whole tasks during a CL scenario. It then trains a model from scratch with the data collected over all tasks at inference time. On the other hand, DFR trains a DNN model with an imbalanced training dataset and re-trains only the last layer with the validation data that is balanced over both class and group labels. Empirical results presented in Prabhu et al. (2020); Kirichenko et al. (2023) show that using a small number of balanced data can be sufficient to achieve comparable performance in terms of CL and debiasing, respectively, compared to more complex methods currently in use.

Following the same principle, we design the BGS algorithm to perform two key steps. First, BGS stores group-class balanced data across all classes and groups encountered in trained tasks in a greedy manner, while concurrently learning a sequence of tasks using any existing CL method. BGS then re-trains only the last layer of the DNN model which has been trained by the CL method, using the group-class balanced exemplar memory. By doing so, we expect that BGS can mitigate the bias of the model akin to DFR while also preventing the forgetting of previous tasks. Notably, BGS is simple and flexible in that BGS is compatible with any existing CL methods without any additional hyperparameters, which makes BGS broadly available as a baseline method. The sampling procedure of BGS is illustrated in Appendix A.

---

[7]In this section, we use "blond_hair" as the class label instead of "young" to create more biased tasks.

Table 1: **The comparison of methods on different CL scenarios.** The average accuracy and BMR or DCA (%) over all tasks are shown. The numbers in parentheses represent the size of the exemplar memory. We bold the results improved by BGS.

(a) Split CIFAR-100S in Task-IL

| Method | Acc. | BMR |
|---|---|---|
| Fine-tuning | 20.47 | 52.86 |
| + BGS (1000) | **30.98** | **46.47** |
| + BGS (2000) | **30.23** | **37.38** |
| LWF | 67.81 | 28.25 |
| + BGS (1000) | **70.46** | **20.60** |
| + BGS (2000) | **70.80** | **17.97** |
| EWC | 41.09 | 46.21 |
| + BGS (1000) | **47.53** | **37.17** |
| + BGS (2000) | **49.79** | **31.89** |
| ER (1000) | 61.27 | 29.19 |
| + BGS | 60.78 | **21.79** |
| ER (2000) | 67.90 | 25.62 |
| + BGS | 66.30 | **18.12** |
| PackNet | 55.01 | 33.75 |
| + BGS (1000) | **55.49** | **31.76** |
| + BGS (2000) | **55.06** | **29.16** |
| LWF + Group DRO | 65.64 | 26.10 |

(b) CelebA[8] in Domain-IL

| Method | Acc. | DCA |
|---|---|---|
| Fine-tuning | 94.56 | 27.00 |
| + BGS (160) | 94.30 | **26.32** |
| + BGS (320) | 94.09 | **24.99** |
| LWF | 95.54 | 29.41 |
| + BGS (160) | 94.87 | **25.45** |
| + BGS (320) | 94.21 | **23.12** |
| EWC | 94.99 | 28.41 |
| + BGS (160) | 88.48 | **15.97** |
| + BGS (320) | 89.66 | **16.65** |
| ER (160) | 94.82 | 27.11 |
| + BGS | 94.51 | **26.42** |
| ER (320) | 94.88 | 27.31 |
| + BGS | 94.67 | **24.78** |
| LWF + Group DRO | 94.26 | 23.67 |

(c) Split ImageNet-100C in Class-IL

| Method | Acc. | BMR |
|---|---|---|
| ER (1000) | 23.49 | 36.53 |
| + BGS | **26.33** | **26.18** |
| ER (2000) | 31.89 | 28.93 |
| + BGS | **33.43** | **21.83** |
| LWF + ER (1000) | 28.03 | 33.60 |
| + BGS | **31.57** | **26.10** |
| LWF + ER (2000) | 34.79 | 29.85 |
| + BGS | **37.68** | **21.84** |
| iCaRL (1000) | 30.15 | 38.30 |
| + BGS | **34.79** | **14.84** |
| iCaRL (2000) | 36.79 | 34.55 |
| + BGS | **41.77** | **12.92** |
| EEIL (1000) | 37.95 | 32.67 |
| + BGS | **37.99** | **22.26** |
| EEIL (2000) | 46.45 | 26.49 |
| + BGS | 43.03 | **17.53** |
| EEIL (1000) + Group DRO | 27.20 | 31.37 |
| EEIL (2000) + Group DRO | 33.94 | 28.78 |

## 6.3 PERFORMANCE COMPARISON

We apply our BGS to standard CL methods with two different sizes of exemplar memory on the three bias-aware CL scenarios. Table 1 presents the performance for naive CL methods, Group DRO combined with the best-performing CL method, and CL methods with BGS plugged in [8]. From the tables, we observe that the naive CL methods exhibit substantial bias in CL models, possibly due to the bias transfer. Naively combining Group DRO with CL methods mitigates the bias of CL models except for Split ImageNet-100C, but it is still sub-optimal in terms of bias metrics when compared to results achieved by incorporating BGS with the same CL method. It could be because of the difficulty of tuning both hyperparameters for a CL method and Group DRO. In contrast, integrating BGS into CL methods consistently yields notable improvements in terms of BMR or DCA across all scenarios, without requiring any additional hyperparameter tuning. Moreover, BGS also enhances average accuracy in most cases on Split CIFAR-100S and Split ImageNet-100C.

In spite of the promising performance of BGS, there remains room for devising a more advanced bias-aware CL method. First, although BGS seeks to remove the influence of spurious features in model predictions by re-training only the last layer in a CL model, the underlying feature representation may still be biased. Namely, BGS may not be a fundamental solution to the issue of bias transfer at its core. Furthermore, when a task contains multiple or continuous types of bias, obtaining the balanced exemplar memory over group labels remains nontrivial. Nevertheless, we emphasize that despite these limitations, our experimental setup and BGS can serve as a crucial stepping stone in developing more advanced bias-aware CL methods.

## 7 CONCLUDING REMARK

With systematic analyses for two-task and multiple-task CL scenarios, we showed the bias can be transferred both forward and backward by typical CL methods that are oblivious to the dataset bias. We pointed out that the bias-aware CL problem is a general and important problem by exhibiting our in-depth analyses and proposed a simple baseline. For future work, we will develop a more principled approach that can accomplish continual learning and debiasing simultaneously.

---

[8]We report the standard deviation and additional results including more recent CL methods in Appendix E.7.

## ACKNOWLEDGMENTS

This work was supported in part by the National Research Foundation of Korea (NRF) grant [No.2021R1A2C2007884] and by Institute of Information & communications Technology Planning & Evaluation (IITP) grants [No.2021-0-01343, No.2021-0-02068, No.2022-0-00113, No.2022-0-00959] funded by the Korean government (MSIT). It was also supported by SNU-LG AI Research Center.

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

## APPENDIX

We offer supplementary materials in this document. Specifically, we provide the detailed algorithm of BGS in Section A. In Section B, we present implementation details including model architecture, optimization, implementation of baseline methods, the range of hyperparameters used, and description for datasets. In Section C, we give the formal definition of DCA. In Section D, we discuss a possible issue with LSF in Domain-IL with further experiments using CelebA[2]. In Section E, we display additional results for accuracy, other various settings, and more baselines. Finally, we add case studies of the bias transfer in more realistic scenarios using satellite images and natural language processing (NLP) datasets.

## A   BGS ALGORITHM

---

**Algorithm 1:** **G**roup-class **B**alanced **G**reedy **S**ampling

**Init** ：Memory $\mathcal{M} = \{\{\}, \dots, \}$ with capacity $K$, labels $\mathcal{L} = \{\}$, count $C = \{0, \dots\}$
**Input** ：A data sample $(x_i, a_i, y_i)$, task id $t$
**Output**：$\mathcal{M}$

1 $k \leftarrow \frac{K}{|\mathcal{L}| \times |\mathcal{A}|}$;
2 **if** $C\big[(a_i, y_i, t)\big] < k$ **then**
3  **if** $\sum_{a \in \mathcal{A}} C\big[(a, y_i, t)\big] == 0$ **then**
4   $\mathcal{L} = \mathcal{L} \cup (y_i, t)$;
5  **end**
6  **if** $\Sigma_{a \in \mathcal{A}, (y,t) \in \mathcal{L}} C\big[(a, y, t)\big] < K$ **then**
7   $\mathcal{M}\big[(a_i, y_i, t)\big] = \mathcal{M}\big[(a_i, y_i, t)\big] \cup (x_i, a_i, y_i, t)$;
8  **else**
9   $(a_j, y_j, t_j) = \text{argmax}_{(a,y,t)}\, C\big[(a, y, t)\big]$;
10   $\mathcal{M}\big[(a_j, y_j, t_j)\big].\text{pop}()$;
11   $\mathcal{M}\big[(a_i, y_i, t)\big] = \mathcal{M}\big[(a_i, y_i, t)\big] \cup (x_i, a_i, y_i, t)$;
12   $C[(a_j, y_j, t_j)] = C[(a_j, y_j, t_j)] - 1$ ;
13  **end**
14  $C\big[(a_i, y_i, t)\big] = C\big[(a_i, y_i, t)\big] + 1$;
15 **end**

---

## B   MORE IMPLEMENTATION DETAILS

### B.1   MODEL ARCHITECTURES AND OPTIMIZATION

For all datasets, we used the AdamW optimizer (Loshchilov & Hutter, 2019) with the following hyperparameters: learning rate of 0.001, weight decay of 0.01, $\beta_1$ of 0.9, $\beta_2$ of 0.999, and $\epsilon$ of $10^{-8}$. For Split CIFAR-100S, we trained ResNet-56 (He et al., 2016) from scratch for 70 epochs using a batch size of 256. For CelebA[2,8] and Split ImageNet-100C, we trained ResNet-18 from scratch for 50 and 70 epochs, respectively, using a batch size of 128. We incorporated the cosine annealing learning rate scheduler, with the maximum number of iterations set to the same as the number of training epochs. We use PyTorch (Paszke et al., 2019) and conduct experiments with servers equipped with AMD Ryzen Threadripper PRO 3975WX CPUs and NVIDIA RTX A5000 GPUs. All experimental results are obtained by averaging the results from 4 different runs.

### B.2   IMPLEMENTATIONS OF CONTINUAL LEARNING METHODS AND BGS

For ER (Chaudhry et al., 2019), iCaRL (Rebuffi et al., 2017), EEIL (Castro et al., 2018), DER (Buzzega et al., 2020), SSIL (Ahn et al., 2021), GDumb (Prabhu et al., 2020), MAS (Aljundi et al., 2018), and DFR (Kirichenko et al., 2023)[9], we implemented the same as their original versions.

---

[9]Results of DER, SSIL, GDumb, MAS, and DFR are reported in Section E.7

Table B.1: **Hyperparameter search ranges.**

| Method | Hyperparameter | Search range |
|---|---|---|
| EWC (Kirkpatrick et al., 2017) | Regularization strength $\lambda$ | $[10^0, 10^9]$ |
| LWF (Li & Hoiem, 2017) | Regularization strength $\lambda$ | $[10^{-4}, 3 \times 10^2]$ |
| ER (Chaudhry et al., 2019) | Memory-to-datasets size ratio $r$ | $[10^{-3}, 10^0]$ |
| iCaRL (Rebuffi et al., 2017) | Memory-to-datasets size ratio $r$ | $[10^{-3}, 10^0]$ |
| EEIL (Castro et al., 2018) | Memory-to-datasets size ratio $r$ | $[10^{-3}, 10^0]$ |
| PackNet (Mallya & Lazebnik, 2018) | Pruning ratio $r$ | $[10^{-1}, 8 \times 10^{-1}]$ |
| Group DRO (Sagawa et al., 2020) | Learning rate of $q$ | $[10^{-8}, 10^0]$ |
| MAS (Masi et al., 2018) | Regularization strength $\lambda$ | $[10^{-4}, 10^1]$ |
| DER (Buzzega et al., 2020) | Regularization strength $\lambda$ | $[10^{-6}, 10^1]$ |

The following describes the CL methods that have been implemented differently from their original versions, in order to improve the practicality, performance, and consistency of our analysis.

In the original EWC algorithm (Kirkpatrick et al., 2017), the snapshot of a CL model should be stored whenever the model is updated from a new task. The snapshots are then used to calculate the importance scores of model parameters in the new task. Namely, the algorithm requires a linearly growing amount of memory to store a sequence of models, which is space-inefficient. To address this issue, we implemented online EWC, proposed in (Schwarz et al., 2018), which averages the importance scores in an online manner without storing a set of models.

For LWF (Li & Hoiem, 2017), we used an average of the distillation losses for each head, not sum of them, to balance between the cross entropy loss and the distillation losses.

For PackNet (Mallya & Lazebnik, 2018), we pruned and re-trained models for 10 epochs with a tenth of the initial learning rate. While the original algorithm of PackNet uses the fixed pruning ratio during learning all tasks in CL, we use different pruning ratios for each task regarding experiments in Sections 4 and 5. Namely, we fix the pruning ratio at the first task to 0.5 and then vary the pruning ratio for the other tasks in order to control the stability and plasticity.

For all rehearsal-based methods, we employed the Reservoir sampling (Vitter, 1985) as a strategy for updating the exemplar memory.

For DFR in CL scnearios, we retrained the last layer with a small subset (20%) of training data of a task after every task.

When BGS is combined with a rehearsal-based method, we do not use an exemplar memory of the rehearsal-based method and adopt only the exemplar memory of BGS. Furthermore, when BGS is applied to iCaRL, re-training with the collected balanced exemplar memory proceeds for the whole layers rather than only for the last layer since iCaRL utilizes the nearest-mean-of-exemplars classification based on the features of exemplars (Rebuffi et al., 2017).

## B.3    HYPERPARAMETER SELECTION FOR EACH RESULT

For each experiment, we evaluated CL methods several times by varying their hyperparameters based on the sets of pre-selected candidates. The candidates are uniformly distributed on a logarithmic scale within a given range, which is reported in Table B.1. For Figure 2 and 3, we discard overlapping results and report the remaining ones among all results for pre-selected hyperparameter candidates.

In Section 4 and 5, we consider the proportion $r$ of the number of training samples of all past tasks as the hyperparameter of rehearsal-based methods with respect to the size of their exemplar memory. To be precise, in Section 4, we store $r\%$ of $T_1$ training samples. For the proportion in Section 5.1 and 5.3, we consider the total number of training samples from $T_1$ to $T_9$ and $T_1$ to $T_2$, respectively.

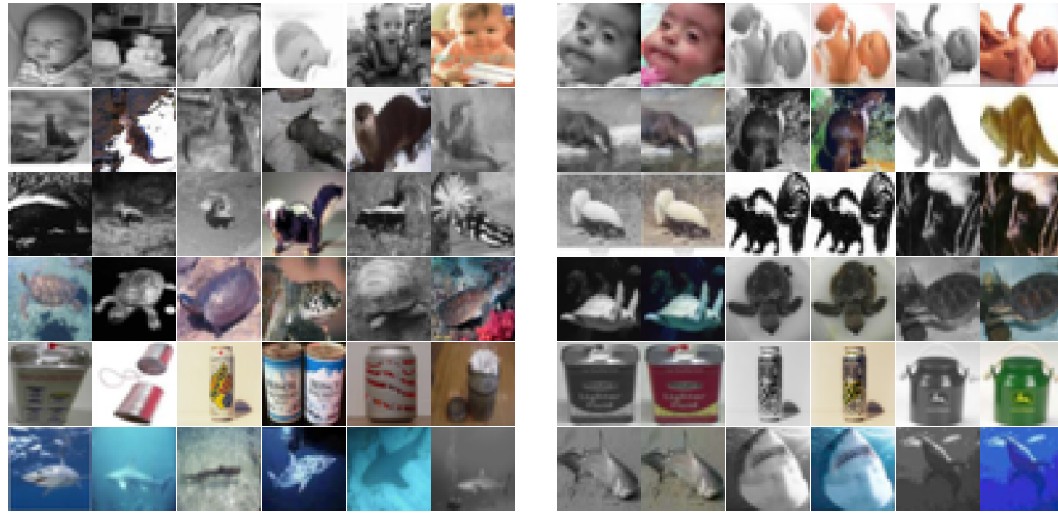

(a) Training samples

(b) Test samples

Figure B.1: **Samples in a certain task with bias level of 2 in Split CIFAR-100S.** Each row represents a specific class within the task. The top three rows represent classes biased toward the grayscale samples, while the bottom three rows contain classes biased toward the color samples. The test dataset includes pairs of images, where each pair contains one grayscale and one color version of the same image.

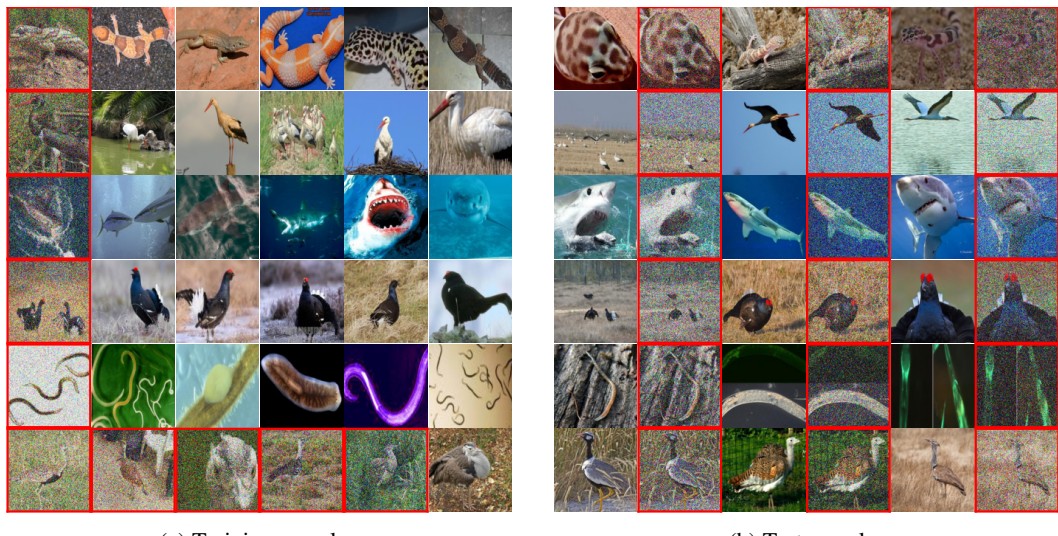

(a) Training samples

(b) Test samples

Figure B.2: **Samples in a certain task with bias level of 0 in Split ImageNet-100C**. Each row represents a specific class within the task. The last row is for the class which is biased towards the Gaussian noise. Red boxes indicate the images with the Gaussian noise. The test dataset includes pairs of images, where each pair contains one original and one noisy version of the same image.

### B.4 DATASETS

**Split CIFAR-100S**. In the training dataset of each task, the first five classes are skewed toward the color group and the latter skewed toward the grayscale group, given the skew ratio. For the test dataset of each task, we have pairs of the same images; ones in color, and ones in grayscale. Figure B.1 illustrates some training and test samples in Split CIFAR-100S. For the experiment in Section 6, the skew-ratio of each task was uniformly randomly sampled from [0.5, 0.99].

**CelebA**. Domain-IL typically assumes the different input distributions with the same label space for each task. To construct such Domain-IL scenario using CelebA, we first select two attributes for group

and class labels, and divide the dataset into several tasks based on other attributes except the two attributes such that each task has different facial attributes, *i.e.*, distinct distributions of non-spurious features. In detail, the two tasks in CelebA$^2$ are divided depending on the "mouth_slightly_open" attribute. For CelebA$^8$ used in Section 5, we adopt "mouth_slightly_open", "black_hair" and "oval_face" attributes to split CelebA into 8 disjoint tasks. In Section 6, we use the "blond_hair" attribute for the class label and "young", "smiling" and "straight_hair" attributes for dividing the CelebA datasets. We note that we use 5,000 samples per each class and group for CelebA$^2$ and subsample the samples with a given skew-ratio $\alpha$, as shown in B.2.

Table B.2: **Dataset construction of CelebA$^2$**

|  | $Y = 0$ | $Y = 1$ ("young") |
|---|---|---|
| $G = 0$ | $5000 \times (1 - \alpha)$ | $5000 \times \alpha$ |
| $G = 1$ ("male") | $5000 \times \alpha$ | $5000 \times (1 - \alpha)$ |

**Split ImageNet-100C**. For this dataset, we assume a specific class at each task is heavily biased toward Gaussian noise. Thus, for each task, we randomly select one class and inject Gaussian noise into $\alpha \geq 1\%$ of the training images of the selected class. For the concrete method to corrupt images, we follow the protocol of Hendrycks & Dietterich (2018) with a noise severity level of 1. The proportions of the noisy images for the other classes are set to 1% (5% in Section 6). Similar to Split CIFAR-100S, we doubled the original test dataset and added Gaussian noise to half of it for the test set. Examples of training and test samples of Split ImageNet-100C are illustrated in Figure B.2a and B.2b. For the experiments in Section 6, $\alpha$ was uniformly randomly sampled from [0.5, 0.95].

## C    FORMAL DEFINITION OF THE DIFFERENCE OF CLASSWISE ACCURACY (DCA)

DCA of a model $h$ at task $T_t$ is defined as follows:

$$\text{DCA}(h) := \frac{1}{|\mathcal{Y}_t|} \sum_{y \in \mathcal{Y}_t} \max_{a, a' \in \mathcal{A}} |\, \text{A}(h, \mathcal{D}_t^{a,y}) - \text{A}(h, \mathcal{D}_t^{a',y})|, \tag{C.1}$$

in which $\mathcal{D}_t^{a,y}$ is the subset of $\mathcal{D}_t$ that are confined to the samples with group label $a$ and class label $y$. DCA indicates the average (across the class) of maximum accuracy gaps between different groups.

## D    ON THE ISSUE OF LOCAL SPURIOUS FEATURES (LSF) IN DOMAIN-IL

In our experiments for Domain-IL in our manuscript, we did not consider the issue of LSF (Lesort, 2022). LSF is a spurious feature in a certain task that can be mitigated when datasets from the whole tasks are collected; for instance, a two-task Domain-IL scenario on CelebA$^2$, in which the skew-ratios of the first and second tasks are $1 - \alpha$ and $\alpha$, respectively. In such a setting, Lesort (2022); Jeon et al. (2023) demonstrate that the rehearsal-based methods can solve this LSF issue because mixing past data in their exemplar memory and current data alleviates the dataset biases, especially when the size of the memory is sufficiently large. Consequently, our claim that a naive CL method promotes bias transfers depending on the relative focus on stability and plasticity may not hold in that case.

However, we claim that rehearsal-based methods could not be a solution for Domain-IL scenarios assuming different distributions of non-spurious features for each task, which Lesort (2022); Jeon et al. (2023) have not addressed. Namely, even if mixing the past and current samples properly to eliminate the spurious correlation between group and class labels, the correlation is still valid when conditioned on the features of which the distribution is shifted across tasks.

To empirically verify this, we additionally consider the six levels of bias from $-6$ to $-1$ by dividing the range of skew-ratio $\alpha$ between the class and group label from $0.01$ to $0.5$ (refer to Table B.2) evenly on a log scale. Then, we carry out a further experiment using two-task scenarios on CelebA$^2$ where the bias levels of $T_1$ and $T_2$ are 6 and -6, respectively. We use two CL scenarios depending on whether there exists a distribution shift of non-spurious features (beyond the bias level change) or not. Specifically, the first scenario assumes that the attribute values for "mouth_slightly_open" in training

Table D.1: **Performance and confusion matrix for each task**. Each value is computed after learning $T_2$ by ER with the maximum size of memory.

(a) Accuracy and DCA for each task

| $T_1$ (bias level: 6) $\rightarrow$ $T_2$ (bias level: -6) | | | | |
|---|---|---|---|---|
| | Dist. shift O | | Dist. shift X | |
| | $T_1$ | $T_2$ | $T_1$ | $T_2$ |
| Accuracy | 95.52 | 88.96 | 79.28 | 75.87 |
| DCA | 63.15 | 44.45 | 3.45 | 3.20 |

(b) Confusion matrix for each group and task in the presence of shift of input distribution

| Group | | Task 1 | | Task 2 | |
|---|---|---|---|---|---|
| | | $\widehat{Y}=0$ | $\widehat{Y}=1$ | $\widehat{Y}=0$ | $\widehat{Y}=1$ |
| G=0 | $Y=0$ | 87 | 163 | 226 | 24 |
| | $Y=1$ | 10 | 240 | 158 | 92 |
| G=1 | $Y=0$ | 241 | 9 | 133 | 117 |
| | $Y=1$ | 172 | 78 | 29 | 221 |

samples of the first and second tasks are different, *i.e.*, 0 and 1, respectively. For the second scenario, there is no specific shift of the attribute, namely, the two tasks are composed of the training samples which are divided uniformly randomly over all attributes (except for the gender bias). Note that both scenarios correspond to the LSF setting because there does not exist a correlation between the class and group labels when the entire training data is collected from the two tasks.

Two tables in D.1 display results for each task after learning $T_2$ using ER with the maximum size of memory. Table D.1a compares accuracy and DCA for each task of the two different CL scenarios. From the results in Table D.1a, we observe that ER does not always mitigate the bias in LSF settings, contrary to what would be expected in (Lesort, 2022). We clearly observe that the bias cancellation only occurs when the input distributions regarding non-spurious features do not shift. We infer that this would be because a CL model can utilize both spurious correlations for each task based on the estimation of different input distributions. The confusion matrices for each group and task in Table D.1b show more clearly that the model relies on the different correlations for prediction in $T_1$ and $T_2$. Namely, for task $T_1$, samples with $G = 0$ (resp. $G = 1$) are more likely to be predicted to be 1 (resp. 0) and vice versa for task $T_2$. In conclusion, the issue of the bias in the LSF settings could not be solved by the typical rehearsal-based CL methods when input distributions regarding non-spurious features are different across tasks, which could also lead to bias transfer to subsequent tasks. In short, although our findings of bias transfer may not hold for settings with LSF based on the same input distributions, we claim that the settings are somewhat non-realistic and our finding could be still valid for the case of LSF with different distributions regarding non-spurious features.

# E ADDITIONAL EXPERIMENTAL RESULTS

## E.1 TWO TASK SCENARIOS WITH DIFFERENT BIAS LEVELS AND NOISE TYPE

Figure E.1 and E.2 display the forward and backward transfer of the color bias on Split CIFAR-100S with two task-CLs. For each figure, we vary the bias levels of $T_2$ (resp. $T_1$) with 2 or 4 and report BMR of $T_1$ (resp. $T_2$) for each CL method and hyperparameter. The results mostly show the analogous trends with the results in Figure 2 and 3. Moreover, it is apparent from the figure that compared to the results in Figure 2 and 3, which are for the bias level of 0 in $T_2$, BMR of $T_2$ is much larger and even the difference in BMR between colored and uncolored points becomes more pronounced. This would be because previously learned biases of a CL model tend to facilitate learning of the dataset bias of the current task more.

In addition, we investigate whether the bias transfer occurs even if the degree of bias of a certain task is much smaller. To that end, we explore the forward (resp. backward transfers) using other two-task CLs with Spliit CIFAR-100S where the bias level of $T_1$ (resp. $T_2$) is 2, which is much milder than the one used in Figure 2(a) (resp. Figure 3(a)). Again, we observe the same trend from E.3 as in Figure 2(a) and 3(a), demonstrating a small amount of bias can be transferred.

Furthermore, in Figure E.4, we report the results for *frost* noise instead of Gaussian noise in the same setting of Split ImageNet-100C. From the Figure, we demonstrate that the bias associated with another type of noise can also be transferred.

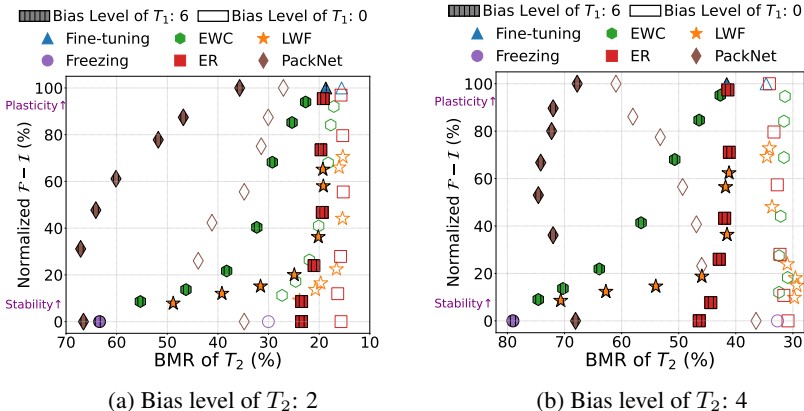

Figure E.1: Forward transfer of bias in two tasks-continual learning on Split CIFAR-100S with different bias levels of $T_2$.

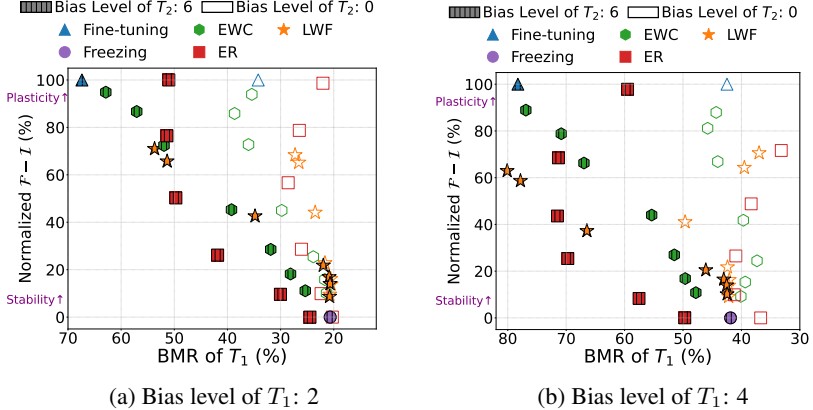

Figure E.2: Backward transfer of bias in two tasks-continual learning on Split CIFAR-100S with different bias levels of $T_1$.

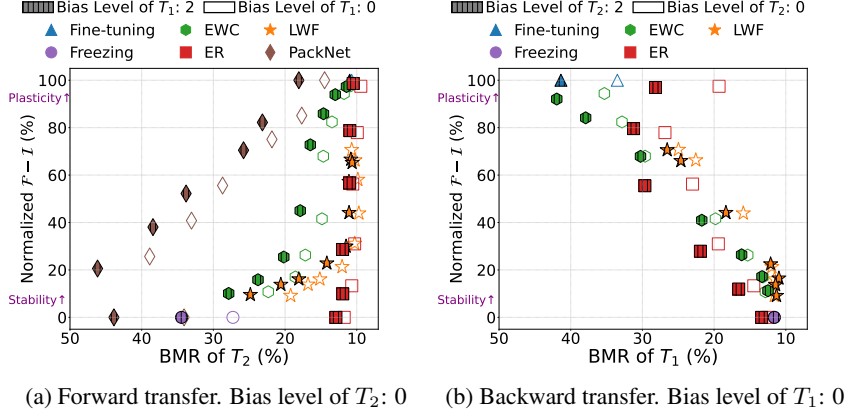

Figure E.3: Effects of bias transfer with mild bias in two tasks-CL on Split CIFAR-100S.

## E.2   ACCURACY FOR CL WITH TWO TASKS

Figure E.5 and E.6 show the accuracy for each point plotted in Figures 2 and 3. Similar accuracy between uncolored and colored points on Split CIFAR-100S and Split ImageNet-100C corroborates

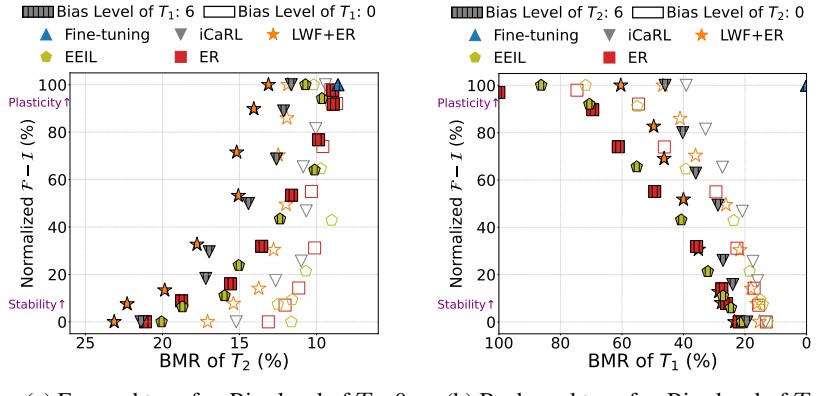

(a) Forward transfer. Bias level of $T_2$: 0      (b) Backward transfer. Bias level of $T_1$: 0

Figure E.4: Transfer of bias in two tasks-continual learning on Split ImageNet-100C with *frost* noise.

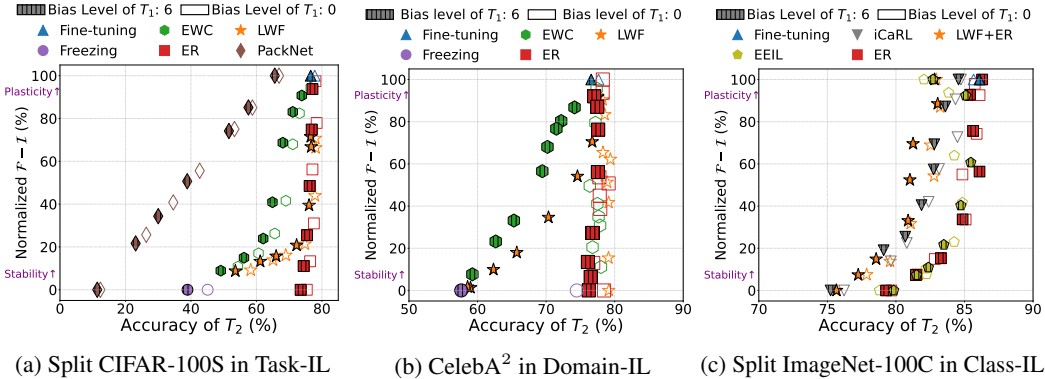

(a) Split CIFAR-100S in Task-IL     (b) CelebA$^2$ in Domain-IL     (c) Split ImageNet-100C in Class-IL

Figure E.5: Accuracy of $T_2$ for the forward transfer of bias in two tasks-continual learning.

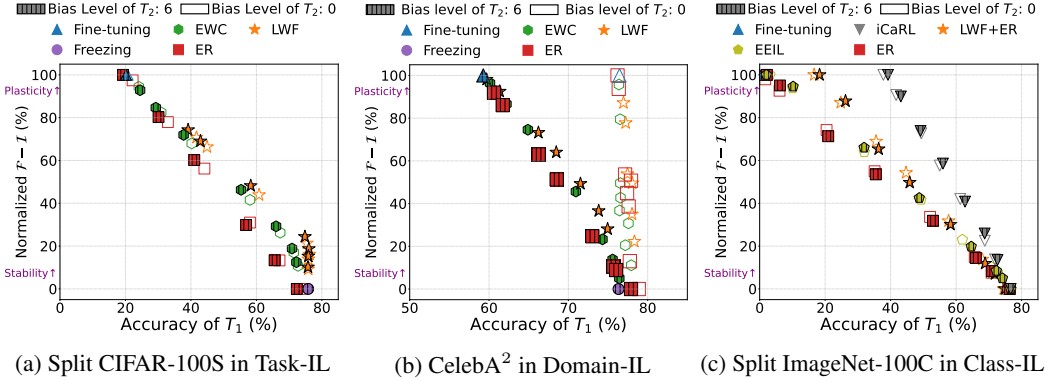

(a) Split CIFAR-100S in Task-IL     (b) CelebA$^2$ in Domain-IL     (c) Split ImageNet-100C in Class-IL

Figure E.6: Accuracy of $T_1$ for the backward transfer of bias in two tasks-continual learning.

that the gaps in BMR under similar Normalized $\mathcal{F} - \mathcal{I}$ in Figures 2 and 3 are not due to differences in accuracy. Furthermore, although the accuracy gaps in CelebA$^2$ have a similar pattern with the DCA gaps in Figures 2b and 3b, the accuracy gaps are smaller in magnitude compared to the DCA gaps.

### E.3 FEATURE REPRESENTATION ANALYSIS FOR BACKWARD TRANSFER OF BIAS

We exhibit the results of CKA analysis for the backward transfer on Split CIFAR-100S. From Figure E.7, it is obviously shown that the CKA value decreases as the regularization strength decreases and

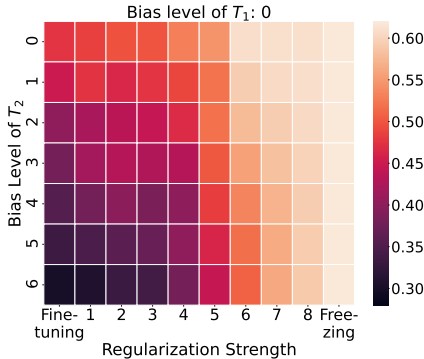

Figure E.7: **CKA on Split CIFAR-100S.** The CKA values between color images and grayscale images in $T_1$ are shown. Each value is calculated after learning $T_2$ by EWC.

the bias level of $T_2$ increases. This again demonstrates that when a CL method focuses on learning a biased current task, *i.e.*, plasticity, the backward transfer of bias by a CL method becomes more obvious.

### E.4 RESULTS FOR ACCURACY WITH A LONGER SEQUENCE OF TASKS

Figures E.8 and E.9a display accuracy corresponding to each point plotted in Figures 5 and 6. It is observed from the figures that the accuracy in the same intervals is roughly the same, so we infer that the gaps of BMR shown in Figures 5 and 6 are due to the bias transfers, not due to the accuracy gaps.

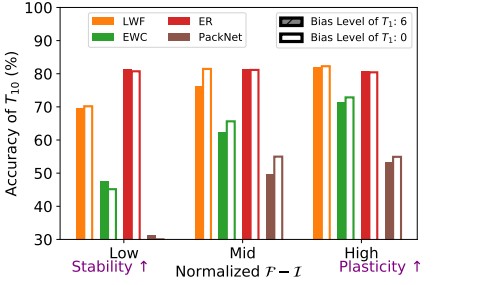
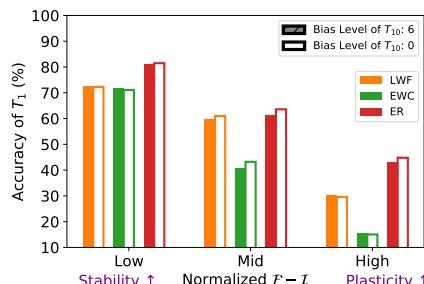

Figure E.8: **Accuracy in longer sequences of Split CIFAR-100S.** The experimental settings in the two plots are the same as in Figure 5a and 5b, respectively.

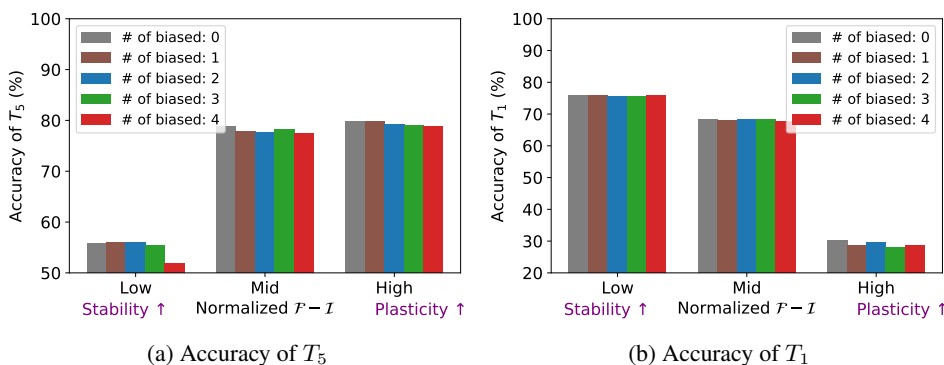

(a) Accuracy of $T_5$         (b) Accuracy of $T_1$

Figure E.9: **Accuracy depending on the number of biased tasks.** The experimental settings are the same as in Figure 6.

### E.5 ACCUMULATION OF THE SAME TYPE OF DATASET BIAS BY BACKWARD TRANSFER

We investigate whether the same type of dataset bias for each task can be accumulated by a CL method by backward transfer. For this experiment, we randomly selected some of the four tasks from $T_2$ to $T_5$ and changed the bias levels of them to 4, *i.e.*, make them more biased. The bias levels of the remained tasks are set to 0. Figures E.10 and E.9b show the BMR and accuracy of $T_1$ depending on the varying number of biased tasks when LWF is applied. Again, we observed the BMR of $T_1$ becomes higher as the number of biased tasks increases especially when CL focuses on high plasticity. This demonstrates that accumulation can happen also backward.

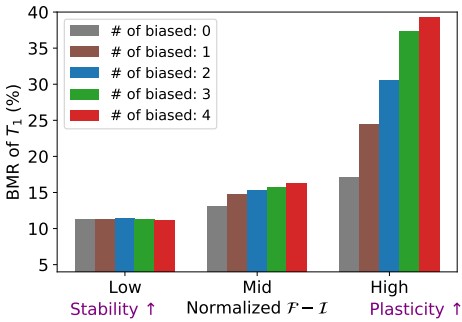

Figure E.10: Accumulation of the same type of dataset bias on Split CIFAR-100S by backward transfer.

### E.6 ACCUMULATION OF THE DIFFERENT TYPES OF DATASET BIAS BY BACKWARD TRANSFER

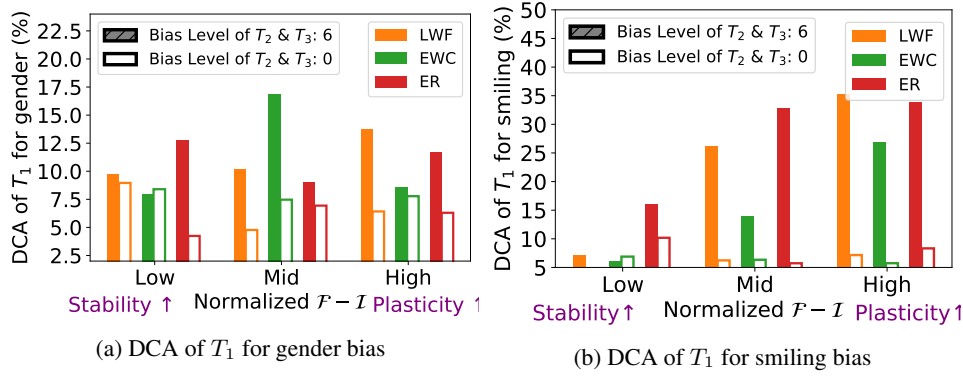

(a) DCA of $T_1$ for gender bias

(b) DCA of $T_1$ for smiling bias

Figure E.11: Accumulation of the different types of bias on CelebA[8] by backward transfer.

We consider sequences of three tasks, similar to Figure 7, to show the different types of dataset bias that can also be accumulated by backward transfer. We assume spurious correlations regarding "gender" and "smiling" attributes for $T_2$ and $T_3$ as "gender" and "smiling" attributes, and vary the bias level of the two tasks while fixing both types of the bias levels of $T_1$ to 0. We again confirm from Figure E.11, that the accumulation of the two types of bias similarly occurs by backward transfer. Note that backward transfer for the bias of the "gender" attribute can happen more in the low or mid regime than the high regime because the knowledge for the bias picked up obtained from the second task can be forgotten after learning $T_3$ in the high regime.

### E.7 PERFORMANCE COMPARISON WITH MORE BASELINES AND AN ADDITIONAL SETTING

In Table E.1, we report the standard deviation for the results in Table 1. We also added results for GDumb (Prabhu et al., 2020), DFR (Kirichenko et al., 2023), and more recent CL methods: MAS (Aljundi et al., 2018), DER (Buzzega et al., 2020) and SSIL (Ahn et al., 2021). Additionally,

Table E.1: **The comparison of methods on different CL scenarios.** The average accuracy and BMR or DCA (%) over all tasks are shown. The numbers in parentheses represent the size of the exemplar memory. We bold the results improved by BGS.

(a) Split CIFAR-100S in Task-IL

| Method | Acc. | BMR |
|---|---|---|
| Fine-tuning | 20.47±1.29 | 52.86±11.15 |
| + BGS (1000) | **30.98±1.91** | **46.47±7.35** |
| + BGS (2000) | **40.23±0.96** | **37.38±6.39** |
| LWF | 67.81±1.00 | 28.25±2.35 |
| + BGS (1000) | **70.46±0.61** | **20.60±1.04** |
| + BGS (2000) | **70.80±0.40** | **17.97±0.44** |
| EWC | 41.09±2.54 | 46.21±2.08 |
| + BGS (1000) | **47.53±1.03** | **37.17±2.24** |
| + BGS (2000) | **49.79±0.79** | **31.89±2.28** |
| ER (1000) | 61.27±0.63 | 29.19±3.43 |
| + BGS | 60.78±0.63 | **21.79±2.22** |
| ER (2000) | 67.90±0.96 | 25.62±1.40 |
| + BGS | 66.30±0.46 | **18.12±1.32** |
| PackNet | 55.01±0.56 | 33.75±5.86 |
| + BGS (1000) | **55.49±0.90** | **31.76±3.01** |
| + BGS (2000) | **55.06±1.25** | **29.16±2.33** |
| MAS | 50.61±0.72 | 41.06±6.69 |
| + BGS (1000) | 49.97±0.51 | **34.06±4.59** |
| + BGS (2000) | 49.23±0.91 | **31.72±3.15** |
| DER (1000) | 50.64±0.17 | 39.16±6.27 |
| + BGS | 48.15±1.03 | **32.50±3.64** |
| DER (2000) | 51.66±0.57 | 37.78±6.34 |
| + BGS | 48.65±0.72 | **30.05±2.90** |
| GDumb (1000) | 36.11±1.70 | 35.36±9.83 |
| GDumb (2000) | 43.70±0.57 | 42.57±10.40 |
| DFR | 20.38±1.44 | 55.01±11.00 |
| LWF + Group DRO | 65.64±0.67 | 26.10±2.47 |

(b) CelebA[8] in Domain-IL

| Method | Acc. | DCA |
|---|---|---|
| Fine-tuning | 94.56±0.42 | 27.00±2.62 |
| + BGS (160) | 94.30±0.26 | **26.32±1.53** |
| + BGS (320) | 94.09±0.60 | **24.99±1.99** |
| LWF | 95.54±0.09 | 29.41±1.60 |
| + BGS (160) | 94.87±0.48 | **25.45±3.12** |
| + BGS (320) | 94.21±0.73 | **23.12±3.16** |
| EWC | 94.99±0.18 | 28.41±1.44 |
| + BGS (160) | 88.48±3.35 | **15.97±3.83** |
| + BGS (320) | 89.66±1.54 | **16.65±2.39** |
| ER (160) | 94.82±0.30 | 27.11±1.64 |
| + BGS | 94.51±0.11 | **26.42±1.55** |
| ER (320) | 94.88±0.13 | 27.31±1.58 |
| + BGS | 94.67±0.06 | **24.78±1.39** |
| MAS | 95.07±0.24 | 26.83±1.68 |
| + BGS (160) | 91.16±2.05 | **18.62±2.67** |
| + BGS (320) | 90.92±0.95 | **17.43±1.72** |
| DER (160) | 94.54±0.15 | 27.19±2.06 |
| + BGS | 94.31±0.38 | **25.99±2.05** |
| DER (320) | 94.51±0.42 | 26.28±1.64 |
| + BGS | **94.76±0.15** | 28.16±2.92 |
| GDumb (160) | 84.75±1.68 | 17.47±2.04 |
| GDumb (320) | 85.71±0.70 | 18.31±1.14 |
| DFR | 93.63±0.94 | 23.34±2.11 |
| LWF + Group DRO | 94.26±0.35 | 23.67±2.23 |

(c) Split ImageNet-100C in Class-IL

| Method | Acc. | BMR |
|---|---|---|
| ER (1000) | 23.49±0.67 | 36.53±1.77 |
| + BGS | **26.33±0.51** | **26.18±1.48** |
| ER (2000) | 31.89±1.32 | 28.93±0.72 |
| + BGS | **33.43±1.12** | **21.83±0.22** |
| LWF + ER (1000) | 28.03±1.28 | 33.60±2.41 |
| + BGS | **31.57±1.59** | **26.10±2.21** |
| LWF + ER (2000) | 34.79±1.12 | 29.85±1.55 |
| + BGS | **37.68±1.19** | **21.84±1.28** |
| iCaRL (1000) | 30.15±0.55 | 38.30±2.96 |
| + BGS | **34.79±0.32** | **14.84±1.43** |
| iCaRL (2000) | 36.79±1.10 | 34.55±1.61 |
| + BGS | **41.77±0.75** | **12.92±0.33** |
| EEIL (1000) | 37.95±0.17 | 32.67±2.70 |
| + BGS | **37.99±1.20** | **22.26±1.21** |
| EEIL (2000) | 46.45±1.30 | 26.49±1.26 |
| + BGS | 43.03±1.02 | **17.53±0.37** |
| SSIL (1000) | 43.76±0.31 | 32.25±3.37 |
| + BGS | 42.56±0.69 | **21.05±1.24** |
| SSIL (2000) | 45.73±0.41 | 31.04±2.56 |
| + BGS | 44.12±0.50 | **19.22±1.28** |
| DER (1000) | 37.15±0.66 | 33.11±2.13 |
| + BGS | 37.10±0.88 | **25.23±0.79** |
| DER (2000) | 47.20±0.82 | 31.90±3.56 |
| + BGS | 45.62±1.15 | **19.33±0.79** |
| GDumb (1000) | 12.88±0.28 | 16.30±4.88 |
| GDumb (2000) | 21.26±0.56 | 20.28±1.65 |
| EEIL (1000) + Group DRO | 27.20±1.45 | 31.37±3.23 |
| EEIL (2000) + Group DRO | 33.94±1.03 | 28.78±2.84 |

(d) Split CIFAR-100S in Class-IL

| Method | Acc. | BMR |
|---|---|---|
| ER (1000) | 18.94±1.17 | 50.88±3.15 |
| + BGS | **20.67±0.76** | **40.87±1.33** |
| ER (2000) | 25.74±0.99 | 47.24±0.41 |
| + BGS | **26.85±0.73** | **26.85±0.73** |
| LWF + ER (1000) | 21.52±1.30 | 51.63±1.48 |
| + BGS | **24.82±1.23** | **41.26±1.13** |
| LWF + ER (2000) | 27.29±1.33 | 46.82±2.21 |
| + BGS | **30.28±0.73** | **30.28±0.73** |
| iCaRL (1000) | 21.29±0.52 | 49.72±4.53 |
| + BGS | **25.36±1.34** | **31.65±2.78** |
| iCaRL (2000) | 28.26±0.84 | 44.05±2.61 |
| + BGS | **32.42±1.18** | **25.40±2.56** |
| EEIL (1000) | 28.04±0.59 | 43.58±1.56 |
| + BGS | **35.19±0.53** | **40.37±0.72** |
| EEIL (2000) | 27.41±0.49 | 35.57±2.52 |
| + BGS | **33.03±0.90** | **30.83±0.38** |
| SSIL (1000) | 27.95±2.03 | 49.39±3.77 |
| + BGS | **31.37±0.46** | **41.49±2.23** |
| SSIL (2000) | 29.30±2.13 | 48.18±2.89 |
| + BGS | **32.26±0.81** | **38.25±2.19** |
| DER (1000) | 26.08±0.91 | 53.87±1.52 |
| + BGS | **31.05±0.64** | **42.72±1.07** |
| DER (2000) | 29.19±0.76 | 52.02±3.20 |
| + BGS | **34.38±0.75** | **37.35±2.28** |
| GDumb (1000) | 12.62±0.62 | 64.56±3.61 |
| GDumb (2000) | 19.24±1.36 | 58.32±4.71 |
| EEIL (1000) + Group DRO | 22.51±0.95 | 40.56±1.79 |
| EEIL (2000) + Group DRO | 27.75±0.80 | 38.28±2.66 |

we compare the baselines in the Class-IL setting of Split CIFAR-100S to provide a more unified evaluation based on Split CIFAR-100S.

# F  ANALYSIS FOR BIAS TRANSFER USING MORE REALISTIC CL SCENARIOS

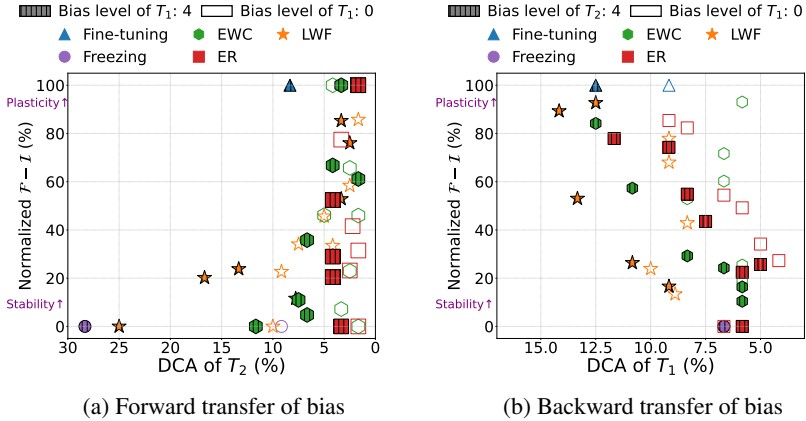

(a) Forward transfer of bias

(b) Backward transfer of bias

Figure F.1: Bias transfer in two tasks-CL (Domain-IL) on FMoW-WILDS dataset.

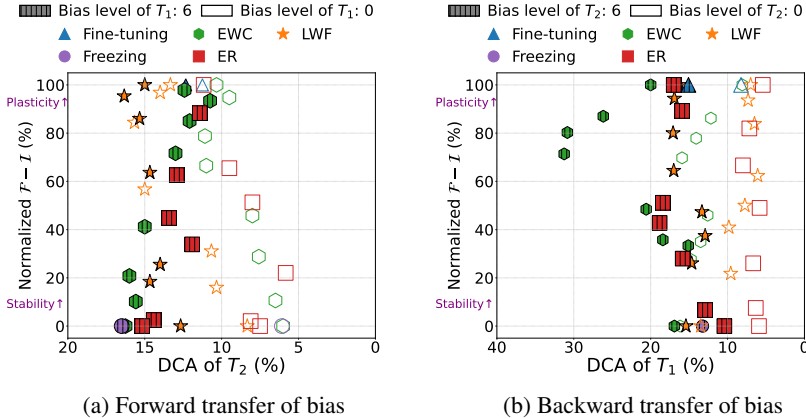

(a) Forward transfer of bias

(b) Backward transfer of bias

Figure F.2: Bias transfer in two tasks-CL (Domain-IL) on NLP datasets.

To extend our findings of bias transfer to more realistic CL scenarios, we constructed two types of two-task CL scenarios: the first one is made from FMoW-WILDS dataset (Koh et al., 2021) and the other one consists of two types of NLP dataset, *i.e.*, MultiNLI (Williams et al., 2017) and WaNLI (Liu et al., 2022).

FMoW-WILDS dataset consists of 141K RGB satellite images annotated with building or land use categories, the year the images were taken, and their geographical regions. Koh et al. (2021) exhibited that a classifier trained from the FMoW-WILDS dataset only performs well on data-rich regions due to the disparities in the number of available data between regions. We assume two-task CL scenarios in time order, where the two tasks consist of subsets of images taken in 2010 and 2016-2017, respectively. We set the binary group label with two regions, *America* and *Asia*, and the binary class label with two categories, *recreational facility* and *toll booth*, which are spuriously correlated with the group label.

We consider another two-task CL scenario where a model is trained with two kinds of NLP datasets, MultiNLI and WaNLI, sequentially. The two datasets are natural language datasets where the task is to determine if a given hypothesis is entailed by, neutral with, or contradicts a given premise. Sagawa et al. (2020) represents that MultiNLI dataset possesses a dataset bias; among premise-hypothesis pairs annotated with the contradiction, most hypothesis sentences contain negation words such as "never" or "no". Based on the findings, we set the group label to the presence of negation words in the hypothesis such that the class labels in each task can be spuriously correlated with group labels.

The figures F.1 and F.2 display how much the dataset bias at the first or second task adversely influences the other task depending on the relative weight on stability and plasticity. The results indicate a similar tendency with 2 and 3. Specifically, we observe again that the forward and backward transfer of bias exists and is significant, and its degree is influenced by the relative focus on stability and plasticity.

