# OpenReview forum: "Continual Learning in the Presence of Spurious Correlations: Analyses and a Simple Baseline"
_ICLR.cc/2024/Conference — ICLR 2024 poster_

### Official Review · Reviewer_C1FY · 2023-10-31

**Soundness:** 1 poor
**Presentation:** 3 good
**Contribution:** 2 fair
**Rating:** 5
**Confidence:** 3

**Summary:**

The paper investigates the impacts of spurious correlations in the continual learning setting.

First, the paper defines the experimental setups and evaluation metrics for investigating the issue of bias transfer.

Then, the paper conducts experiments, arguing that bias exists and existing CL approaches cannot handle the bias. This is done first for the case of two tasks, and then generalized to longer task sequences.

Next, the paper proposes a method to address the bias. The method consists of retraining the last linear layer using a balanced set of data samples.

**Strengths:**

Omitted.

**Weaknesses:**

In my humble opinion, the paper has several weaknesses:

- The paper has limited novelty. In particular, the proposed problem is a basic combination of continual learning and bias of machine learning models. The proposed approach is a direct variant of Kirichenko et al. (2023). An analogy would be that the paper derives a corollary from a theorem in prior work.
- In the experimental setting, the paper changes the input data by skewing them towards gray-scale images (or colored) images. This creates several issues:
  - This setup makes the experiments artificial, human-made, and synthetic. The practical relevance is to be evidenced.
  - The notation of bias is subjective. The way the paper changes the input data is just to make a shift of distribution or create subclasses from existing classes, rather than creating any bias. Retraining the last layer using a balanced set of data would of course improve in this case as it reduces the performance gap between (sub)classes and improves DCA. Therefore, the story the paper tries to convey is in my eyes a word game that is not convincing.

**Questions:**

I have no specific questions

---

> ### Author Response · Authors · 2023-11-19
> **Rebuttal by authors**
>
> We thank Reviewer C1FY for valuable feedback. The followings are our replies to the comments.
>
> **Weakness**
>
> > W1. novelty.
>
> Thank you for your comment, but we contend that while our problem setting involves the combination of the two topics, our key findings do not solely derive from the mere consideration of the two topics simultaneously. Specifically, most existing works on dataset bias only focus on offline learning settings. Conversely, most CL works overlook the issue of bias in CL. Consequently, there remain unpredictable findings in bias-aware CL scenarios, such as whether some factors may make dataset bias of a certain task in CL more severe or if applying existing CL and debiasing methods adequately addresses bias issues in CL.
>
> Our paper conducts extensive experimental analysis to explore these considerations. We offer substantial experimental evidence in [Section 4,5] showing the bias transfer by typical CL methods. Moreover, the results for performance comparison in [Section 6] show the ineffectiveness of typical CL methods or naive combinations between CL and debiasing methods in bias-aware CL settings. At this point, our key findings provide a novel and significant perspective for many CL research communities.
>
> Together with our empirical findings, we offer a standardized evaluation protocol as a tool for developing more advanced bias-aware CL methods. We emphasize that we propose BGS to serve as a baseline method when assessing a new bias-aware method using our protocol. Hence, for BGS, we argue that prioritizing simple implementation, performance and flexibility may be more crucial than pursuing novelty in method design.
>
> > W2-1. synthetic.
>
> Skewing vision benchmark datasets (e.g., CIFAR, MNIST) with color is a conventional way employed in numerous debiasing studies [A1, A2]. Following these prior works, we constructed Split CIFAR-100S. While it seems to be quite synthetic, we argue that we can obtain valuable insights from experimental results on Split CIFAR-100S. Furthermore, to complement these synthetic settings, we additionally use CelebA, FMoW, and NLP datasets that contain a realistic gender bias (for results on the two latter datasets, refer to replies for Reviewer UUYC and 6w81, respectively). Hence, we assert that our synthetic color bias setting should not be perceived as a weakness, rather, it can diversify and enrich our experimental results.
>
> > W2-2. subjectiveness of bias
>
> We skewed each task of Split-CIFAR100S with color, such that the training dataset of each task predominantly consists of gray images for the first five classes and color images for the remaining five classes. Namely, we create color bias, that is, spurious correlation between group and class labels. As a result, a biased model trained from the color-skewed training dataset would tend to predict gray images as one of the first five classes and vice versa. This shows our bias setting for Split-CIFAR100S mimics real-world spurious correlation that exists such as in face recognition or toxicity classification.
>
> **Question**
> N/A
>
> **References**
> [A1] Wang et al., Towards Fairness in Visual Recognition: Effective Strategies for Bias Mitigation, CVPR, 2020.
> [A2] Nam et al., Learning from Failure: Training Debiased Classifier from Biased Classifier, NeurIPS, 2020.

---

> > ### Author Response · Authors · 2023-11-23
> > **Any feedback on the rebuttal?**
> >
> > We again thank reviewer C1FY for the detailed and valuable comments. We tried our best to respond to all the concerns, and we hope that our rebuttal effectively addressed your concerns and clarified the intricacies of our study. Would you please let us know whether our rebuttal helped the reviewers re-assess the paper? Please let us know whether there is anything unclear in our rebuttal.
> >
> > Thank you very much in advance.

---

### Official Review · Reviewer_A8St · 2023-11-01

**Soundness:** 3 good
**Presentation:** 3 good
**Contribution:** 3 good
**Rating:** 6
**Confidence:** 3

**Summary:**

This paper investigates the impact of spurious correlations in the CL setting. Through comprehensive experiments, it confirms the existence of bias transfer in CL, affecting model predictions in both forward and backward directions. Then, they establish standard experimental settings, bias-aware CL scenarios, and evaluation protocols and introduce a practical baseline method called "Group-class Balanced Greedy Sampling (BGS)" for advancing bias-aware CL techniques.

**Strengths:**

1.    This article has a clear motivation. I also agree that investigating spurious correlations is a highly worthwhile topic in the CL setting.
2.    The related work section of the article provides an excellent summary of the relationship between CL methods and spurious correlations.
3.    The paper's approach to investigating bias awareness from three distinct angles, i.e., model bias, the relative focus on the Stability-Plasticity trade-off, and bias transfer, is comprehensive and convincing.

**Weaknesses:**

1.    The readability of the paper could be improved. The abstract and introduction should be revised to provide a more engaging and clearer overview of the research.
2.    The paper introduces the Bias-flipped Mis-classification Rate (BMR) and the Difference of Classwise Accuracy (DCA) as metrics, but it lacks a detailed comparison of these metrics on the proposed benchmark datasets. It would be valuable to provide an in-depth analysis of how these metrics perform under different scenarios.
3.    Table 1 should include further comparisons of CL baselines. A more comprehensive analysis of the performance of other CL baselines on the proposed benchmark datasets would provide a stronger basis for evaluating the proposed method.
4.    While this paper introduces the BALANCED GREEDY SAMPLING (BGS) method, its novelty appears to be limited. The paper could benefit from a more thorough exploration of innovative techniques in the bias-aware CL domain.
5.    The experimental analysis and the overall structure of the paper should be enhanced. This paper reads more like a forward-looking exploration of bias-aware scenarios rather than a comprehensive research work. It would be beneficial to present a more detailed and rigorous experimental analysis.

**Questions:**

1.    Can the paper provide a more detailed comparison between its proposed methods BGS and the GDumb and DFR CL methods to highlight their differences in addressing bias-aware CL scenarios?
2.    Does Table 1 in the paper, which indicates a significant improvement in bias-aware CL scenarios when using the BGS method, imply that BGS may be less robust in scenarios with larger datasets, as increasing data has a limited impact on model improvement in all three scenarios?

---

> ### Author Response · Authors · 2023-11-19
> **Rebuttal by authors**
>
> We thank Reviewer A8St for the positive comments regarding the motivation of our research and various perspectives. We also thank you for addressing the limitations of our experimental settings and method. The followings are our replies to the comments.
>
> **Weakness**
>
> > W1
>
> We feel confident that we have presented a clear overview of the paper in both the abstract and the introduction. Would you kindly give us your feedback on any specific sentences that may be unclear, or more concrete ways for how we improve the clarity/engagement of our abstract and introduction further? We will be eager to reply to any concrete comments, but it is very hard for us to reply to the current comment.
>
> >W2
>
> We appreciate your feedback on the points we have overlooked. In [Section 3.1], we clarified our use of DCA as a surrogate metric for BMR in cases where BMR is unavailable. To elaborate further, we define the degree of bias in a model as its sensitivity to group attributes in predictions of the model. Ideally, BMR serves as an optimal bias metric by comparing predictions between original and bias-flipped samples. However, if such counterfactual samples are unavailable, as in CelebA dataset, we indirectly gauge the degree of bias in the model by examining the accuracy difference between bias-aligned and bias-conflicted groups, i.e., DCA.
>
> However, it is important to note that DCA can be somewhat misleading as a bias metric, especially in some extreme cases in which the overall accuracy is very low. For instance, a completely random classifier, with identical accuracy for each group (1 over the number of classes), results in a DCA of 0, but the model may indeed have bias so that the resulting BMR can be high. Therefore, it is preferable to use BMR as the primary bias metric.
>
>
> The table below compares BMR and DCA for the results on Split CIFAR-100S, reported in [Table 1, Section 6]. We observe that DCA results mostly have a similar trend to BMR results, suggesting that DCA can serve as the surrogate metric of BMR. However, we note that Fine-tuning results in very low accuracy, hence, the DCA becomes unexpectedly low while the corresponding BMR is high. Moreover, DCA even worsens when BGS is applied due to the slight accuracy improvement, while BMR shows improvement. This result shows that DCA can sometimes be misleading as a bias metric. We will add this discussion in the Appendix of the final version.
>
> |     | Acc. | BMR  | DCA  |
> | --- | ---- | ---- | ---- |
> | Fine-tuning | 20.47 | 52.86 | _11.95_ |
> | + BGS (1000) | 30.98 | 46.47 | _14.36_ |
> | LWF    | 67.81 | 28.25 | 25.91 |
> | + BGS (1000) | 70.46 | 20.60 | 17.54 |
> | EWC | 41.09 | 46.21 | 22.96 |
> | + BGS (1000) | 47.53 | 37.17 | 18.54 |
> | ER (1000) | 61.27 | 29.19 | 21.74 |
> | + BGS | 60.78 | 21.79 | 12.94 |
> | PackNet | 55.01 | 33.75 | 21.89 |
> | + BGS (1000) | 55.49 | 31.76 | 18.89 |
> | LWF + Group DRO | 65.64 | 26.10 | 22.84 |
>
> >W3
>
> We have already reported the results of three additional methods, GDumb, MAS, and DER in [Appendix E], which are more recently developed CL methods. We would greatly appreciate your more concrete feedback on additional baselines to add.
>
> >W4
>
> Our main contribution lies in highlighting the issue of dataset bias in CL, rather than introducing a new state-of-the-art bias-aware CL method, because many CL researchers may have inadvertently overlooked this critical issue. To that end, we thoroughly demonstrated that the bias transfer indeed exists and is significant in CL, emphasizing the importance of addressing the bias transfer issue in CL. We then offered a standardized evaluation protocol and BGS for foundational tools for developing new bias-aware CL methods. In this context, we believe that prioritizing simplicity, flexibility and efficiency of BGS is more essential than pursuing novelty, as the development of a highly innovative method is beyond the scope of our paper.
>
> >W5
>
> We believe that we provided *in-depth* analytical results for the issue of bias in CL beyond naively exploring it. Specifically, we showed that in various bias-aware CL scenarios, bias transfer exists and is influenced by Normalized $\mathcal{F-I}$ values by analyzing the prediction and representation of CL models. We do not believe a paper is only worthy of publication if it surprises. While the intuition of spurious correlations affecting continual learning may have existed to some extent, we believe our paper shows substantial evidence of the idea via thorough and systematic experiments and analyses.

---

> > ### Author Response · Authors · 2023-11-19
> > **Rebuttal by authors**
> >
> > **Question**
> > >Q1
> >
> > As we mentioned in Section 6.2, GDumb and DFR are only designed as a CL and a debiasing method, respectively. Namely, GDumb collects data samples into an exemplar memory by considering the balance over class labels. As a result, GDumb uses biased exemplar memory and produces highly biased models (see [Table E.1 in Appendix E]).
> >
> > On the other hand, since DFR is not a CL method, applying DFR for debiasing a certain biased model can lead to catastrophic forgetting of previously learned tasks. To show this empirically, we additionally reported the results for DFR on Split CIFAR-100S and CelebA$^8$ in [Table E.1, Appendix E].
> >
> > >Q2
> >
> > Could you kindly elaborate on the specific aspects of our findings that indicate a constrained influence of increased data? From our perspective, [Table 1, Section 6] shows that as the size of exemplar memory gets larger, BGS can be more effective in improving average accuracy and bias metrics.

---

> > > ### Author Response · Authors · 2023-11-23
> > > **Any feedback on the rebuttal?**
> > >
> > > We again thank reviewer A8St for the detailed and valuable comments. We tried our best to respond to all the concerns, and we hope that our rebuttal effectively addressed your concerns and clarified the intricacies of our study. Would you please let us know whether our rebuttal helped the reviewers re-assess the paper? Please let us know whether there is anything unclear in our rebuttal.
> > >
> > > Thank you very much in advance.

---

> > > > ### Comment · Reviewer_A8St · 2023-11-23
> > > >
> > > > Thanks for the detailed explanation and experimental results! Some of my concerns are addressed well. I will raise my score.

---

> > > > > ### Author Response · Authors · 2023-11-23
> > > > > **Official comment by authors**
> > > > >
> > > > > Thank you very much for your positive and helpful feedback!

---

### Official Review · Reviewer_6w81 · 2023-11-01

**Soundness:** 3 good
**Presentation:** 3 good
**Contribution:** 3 good
**Rating:** 8
**Confidence:** 4

**Summary:**

The paper revolves around the subject of bias transfer in continual learning (CL). The authors develop an experimental framework examining six CL strategies using two evaluation metrics across three scenarios, ensuring comprehensive analysis of the problem. Their findings show that CL techniques, unaware of dataset bias, can transfer such biases in both forward and backward directions. In response to this issue, they suggest a novel approach, Group-class Balanced Greedy Sampling (BGS), aimed at mitigating bias transfer. This paper is unique in the sense that it deliberates on the existence of spurious correlations in the CL context and calls for attention to develop bias-aware mechanisms in CL.

**Strengths:**

- The authors address an often-overlooked issue of bias transfer in CL and provide a well-motivated argument adopting a fresh perspective in the field of CL.
- Their analysis is fairly extensive, using six CL methods with three different scenarios.
- The novelty of the proposed BGS method to mitigate bias without requiring any additional hyperparameter tuning enhances the paper's contribution.
- The empirical evidence that reveals the existence of bias transfer in CL and its subsequent impact on the tasks makes a significant contribution to the field.

**Weaknesses:**

- While the empirical investigation of CL provides valuable insights, the novelty of findings and proposed BGS approach could be more explicitly addressed considering existing similar work in the domain.
- The main limitation of this paper lies in the diversity of the samples used. The authors base their experiments on three benchmark datasets, all of which are synthetically created for Continual Learning (CL). From personal observation, it has been noted that such models can perform adequately even with a few hundred examples, contradicting the need for the 2000 or 4000 examples that BGS involves. By conducting tests in more authentic scenarios (such as shift in time/linguistic diversity, or the nature of the sequence to sequence task (like going from question answering to machine translation to paraphrasing etc)), a stronger foundation of support for the results could be established.
- The paper's dual research goals are compressed into a limited amount of space, making comprehensive comprehension challenging. More in-depth discussion or a more detailed layout for the proposed BGS method would be beneficial.
- The underlying mechanisms contributing to bias transfer are not entirely delved into in the paper.

**Questions:**

- The role and impact of the stability-plasticity trade-off on bias transfer in CL could be more thoroughly explored.
- The experimental design lacks uniformity across all three CL scenarios and should strive for a standardized evaluation protocol.
- It would be insightful to know if the bias transfer problem exists in substantial real-world applications (as shared in weaknesses) and what limitations exist in the experimental setup.
- Why did you use LWF + Group DRO as a comparison in Tables 1(a) and 1(b) when ER is better performing than LWF? Could you share your rationale here?

---

> ### Author Response · Authors · 2023-11-19
> **Rebuttal by authors**
>
> We thank Reviewer 6w81 for the supportive comments, e.g., “well-motivated and extensive studies for the bias transfer in CL and impactful results”. We also thank you for pointing out the limitations of our experimental settings and method. The followings are our replies to the comments on the weakness.
>
> **Weakness**
>
> We first note that [A1-A4] are all cited in the original manuscript, and we are referring them here again for clarity.
>
> >W1
>
> We have already addressed the key differences between our paper and existing related works throughout the manuscript. As we outlined in [Section 1], [A1] empirically investigated the forward transfer of a bias when a model pre-trained on a biased dataset is fine-tuned with a downstream task. However, their findings do not cover the backward transfer of bias and the _continual_ learning of the task sequences. Furthermore, as stated in [Section 2] and [Appendix D], [A2, A3] pose the issues of dataset bias in CL. Their analysis, however, only focused on the issue of local spurious features (LSF) in Domain-IL and lacked a comprehensive analysis of the transferability of bias.
>
> We believe the above comparison of our work with others, which is already clearly written in the original manuscript, is quite explicit, and it underscores the novelty of our paper. If the reviewer lets us know further concrete questions regarding the novelty of our paper, we will be glad and committed to addressing them in further responses.
>
> > W2
>
> We appreciate your comment on sample diversity. First of all, we would like to clarify a potential misunderstanding regarding our experimental settings in [Section 6]. Specifically, for examplar-based CL methods, including BGS, we used 1000 or 2000 examples, **_not_** *2000 or 4000 samples* as the reviewer has mentioned. These sample quantities fall within the range commonly considered by many CL studies, e.g., [A4].
>
> Meanwhile, to show that our findings of bias transfer are valid even for more realistic datasets, we conducted further experiments on two-task CL scenarios involving a sequence of two kinds of natural language datasets: MultiNLI and WaNLI. The task of both datasets is to determine if a given hypothesis is entailed by, neutral with, or contradicts a given premise. It is widely known that the contradiction class is spuriously correlated with the presence of negation words in hypothesis sentences. Thus, we set the group label as the presence of negation in the sentence. We then control the skewness between the class label and group label to investigate the bias transfer.
>
> The results, as depicted in [Figure F.2, Appendix F] in the revised version, indicate that the bias transfer is evident even in more realistic CL scenarios based on NLP datasets. Additionally, it is clearly shown that the degree of bias transfer is influenced by Normalized $\mathcal{F}-\mathcal{I}$, in line with our findings on other datasets in the original manuscript.
>
> > W3
>
> Our primary contribution lies in highlighting the bias issue in CL. In this context, we have allocated a relatively limited space for explaining the BGS method since our intention to introduce BGS is to offer a sensible and strong baseline for developing a more advanced bias-aware CL method. Furthermore, we believe that we have provided detailed descriptions of the algorithm, performance, and limitations of BGS in [Sections 6.2 and 6.3]. We would greatly appreciate the reviewer for letting us know any specific aspects that need further clarification, and we are fully committed to incorporating them in the further response and the final version.
>
> > W4
>
> Throughout the paper, we believe we have already demonstrated the underlying mechanism of bias transfer --- specifically, we demonstrated that the degree to which a CL method focuses on stability (resp. plasticity) turns out to be the main factor in forward (resp. backward) transfer of bias. That is, if the previous tasks contain some dataset bias, the bias obtained in the previous task is preserved when Normalized $\mathcal{F}-\mathcal{I}$ is low (i.e., when the stability is more prioritized), and it negatively influences future tasks (i.e., the forward transfer of the bias occurs). The mechanism for the backward transfer of bias can be described similarly with the concept of plasticity.
>
>
> **References**
> [A1] Salman et al., When does Bias Transfer in Transfer Learning?, arXiv preprint arXiv:2207.02842, 2022.
>
> [A2] Lesort, Timothée., Spurious Features in Continual Learning. AAAI Bridge Program on Continual Causality, PMLR, 2023.
>
> [A3] Jeon et al., Learning without Prejudices: Continual Unbiased Learning via Benign and Malignant Forgetting, ICLR, 2023.
>
> [A4] Prabhu et al., GDumb: A Simple Approach that Questions Our Progress in Continual Learning, ECCV, 2020.

---

> > ### Author Response · Authors · 2023-11-19
> > **Rebuttal2 by authors**
> >
> > **Question**
> > > Q1
> >
> > Rather than the stability-plasticity trade-off itself being a factor in the bias transfer, focusing on either stability or plasticity is connected to bias forward or backward transfer, respectively. Please see the last reply in the weakness part above.
> >
> > > Q2
> >
> > We believe that in [Section 6], we have offered a standardized experimental protocol. We have maintained consistency by employing the same performance metric and hyperparameter selection rule for each CL scenario. While the bias metrics are different between CelebA and other datasets, we have opted for DCA as a surrogate metric of BMR (for a more in-depth comparison between these two metrics, please refer to the second reply in the weakness part for reviewer A8St). Furthermore, we have established the three CL scenarios, each addressing different categories of CL (i.e., Task-IL, Domain-IL and Class-IL). The reason why CL methods are different for each CL scenario is that applicable CL methods differ for each scenario. However, we believe the diversity in CL categories and methods does not imply an inconsistent evaluation setup but rather contributes to a more general and standardized framework.
> >
> > >Q3
> >
> > For analysis on the issue of bias transfer in real-world CL applications, please refer to our reply in the weakness part.
> >
> > The main limitation of our experimental setup is the requirement for training data annotated with group labels and a sufficient number of samples within bias-conflicted groups to control the bias level of each task. We will mention this more clearly in the final version of the paper.
> >
> > > Q4
> >
> > We selected LWF as it is the best-performing CL method in terms of average accuracy compared to the results of ER when the sizes of exemplar memory are 1000 and 160 in [Table 1(a) and (b), Section 6], respectively.

---

> > > ### Comment · Reviewer_6w81 · 2023-11-22
> > > **Thanks for your thoughtful response**
> > >
> > > I appreciate the time the in-depth response from the authors. The response and the updated paper addresses my concerns. I've updated the score to reflect that.

---

> > > > ### Author Response · Authors · 2023-11-23
> > > > **Official comment by authors**
> > > >
> > > > Thank you very much for your positive comments and helpful feedback!

---

### Official Review · Reviewer_UUYC · 2023-11-08

**Soundness:** 3 good
**Presentation:** 3 good
**Contribution:** 3 good
**Rating:** 6
**Confidence:** 4

**Summary:**

While most continual learning (CL) algorithms focus on the stability-plasticity trade-off, this study highlights the overlooked impact of dataset bias on knowledge transfer. Through systematic experiments on three benchmark datasets, the authors show that standard CL methods can transfer bias between tasks, affecting both forward and backward learning. The degree of bias transfer depends on whether the CL methods prioritize stability or plasticity. Bias transfer accumulates in longer task sequences. To address this issue, the authors propose a standardized experimental setup and introduce a simple yet effective baseline method called Group-class Balanced Greedy Sampling (BGS).

**Strengths:**

- Amongst the works that look at CL and distribution shift, this is the one of the first papers to do an empirical study on impact of dataset bias on three different forms of CL: task-IL, domain-IL and class-IL.

- The authors do a good job of structuring their findings, by first illustrating results in the two task case, followed by multiple tasks in a sequence. The metrics for bias (BMR/DCA) and CL (normalized $\mathcal{F}-\mathcal{I}$) are clearly defined.  In general the presentation is good, and the writing is clear. Particularly, experiment results in section 4 and 5 are well presented with easy to read plots.

- The CKA analysis done on representations of penultimate layer (Sec 4.3) strengthens the empirical results observed.

- The proposed baseline (group-class balanced greedy sampling) is fairly simple and combines ideas from GDumb and DFR. The algorithm presents significantly lower BMR over CL baselines in class-IL and task-IL settings.

**Weaknesses:**

- The datasets used for the experiments are small scale and synthetic. Using more natural datasets that evolve over time, e.g. FMoW dataset from Wilds benchmark, or other datasets from the Time-Wilds paper would be helpful. In some of these datasets, there is also group/attribute information that can be used to measure bias.

- The paper can be improved with experiments that use a pretrained model (like CLIP), and then perform continual learning. It would be interesting to see if the same trends hold, or are they amplified/diminished with respect to forward/backward bias transfer.

- I understand that theoretical analysis is not always feasible or even helpful, but for spurious correlations there exist simple settings/distributions in the SC literature where the SC induces failure even in linear models (see [1, 2, 3]). Extending these frameworks to the CL setting and then proving formal claims about forward/backward bias transfer can make the claims in this paper much stronger and build understanding to develop mitigation strategies and algorithms.

[1] Nagarajan, Vaishnavh, Anders Andreassen, and Behnam Neyshabur. "Understanding the failure modes of out-of-distribution generalization." arXiv preprint arXiv:2010.15775 (2020).

[2] Ghosal, G. R., Setlur, A., Brown, D. S., Dragan, A., & Raghunathan, A. (2023). Contextual Reliability: When Different Features Matter in Different Contexts.

[3] Sagawa, Shiori, et al. "An investigation of why overparameterization exacerbates spurious correlations." International Conference on Machine Learning. PMLR, 2020.

**Questions:**

- In figure 4, for row 1, why does the CKA value drop even when bias of T1 is zero?
- In figure 5, how does PackNet do on backward transfer?
- Also can authors comment on why ER does much better than LWF and PackNet in general?

---

> ### Author Response · Authors · 2023-11-19
> **Rebuttal by authors**
>
> We thank Reviewer UUYC for the positive comments, e.g., “well-motivated and thorough studies for the bias transfer in CL and interesting results”. We also thank you for pointing out the limitations of our experimental settings and method. The followings are our replies to the comments.
>
> **Weakness**
> > W1
>
> Thank you for suggesting to add experiments on real-world datasets. Following the suggestion,  we conducted further experiments on bias transfer for the two-task CL scenarios on the more realistic FMoW dataset. We set the class and group labels as the "building" category and "regions" of images, respectively. We construct two tasks depending on the year in which images were taken (please refer to [Appendix F] in the revised version for a detailed description of our two-task CL scenarios). Our observations from [Figure F.1 in Appendix F] again indicate the consistent trends of the bias transfer. Namely, both the forward and backward transfers of the bias exist and those transfers are affected by the relative focus on plasticity and stability. In the final version, we plan to include additional results for the longer sequences on the FMoW dataset.
>
> > W2
>
> Thank you for your suggestion. Following your suggestion, we carried on additional experiments in two-task CL scenarios when using a pre-trained model. With the same setting as experiments for forward transfer in [Section 3], we compare the two types of CL scenarios; the bias level of the first task is 0 or 6, while the bias level of the second task is fixed as 0. The table below compares BMRs of T$_2$ on Split ImageNet-100C after learning the pre-trained model in the two CL scenarios. Due to the time limit, we reported the results for only two kinds of CL methods (ER and ER+LWF) with only two hyperparameters (the memory sizes corresponding to 10% and 30% of the T$_1$ training dataset). Although normalized $\mathcal{F}-\mathcal{I}$ is not perfectly aligned for each memory size, BMR gaps in the table imply that bias transfer occurs and is affected by normalized $\mathcal{F}-\mathcal{I}$ values even when using a pre-trained model.
>
>
> | CL method | Memory size | Norm. F-I(level 0/6) | BMR T$_2$ (level 0/6) | BMR Gap |
> | -------- | -------- | --- | --- | --- |
> | ER | 10% | 49.56/41.09 | 18.24/22.22 | 3.98 |
> |          | 30% | 19.70/16.88 | 10.86/23.26 | 12.4 |
> | ER+LWF     | 10%     | 47.54/39.97 | 16.21/24.59 | 8.36 |
> |    | 30%     | 15.81/14.77 |18.21/30.66 | 12.45 |
>
> > W3
>
> We sincerely appreciate your constructive feedback. We have comprehensively reviewed the papers you recommended but found that constructing a theoretical analysis on the bias transfer in continual learning seems very challenging, particularly for the short rebuttal period. The main reason for such a challenge is that we currently do not even have a rigorous theory on continual learning or knowledge transfer. Consequently, we decided to defer the theoretical analyses on our work to future work, but we will definitely pursue the direction after finalizing the current paper.
>
> **Question**
> > Q1
>
> Thank you for the detailed question. We note that even if the bias level is set to 0 (i.e., the skew-ratio of the dataset is 0.5), the BMR could still be non-zero --- in fact, we observed that BMR for a model trained on a Split CIFAR-100S dataset with bias level of 0 was 11%. This would be the reason for the bias transfer occurring in row 1 of [Figure 4, Section 4].
>
>
> > Q2
>
> As we mentioned in [Section 4.2], for PackNet, the backward transfer does not occur since it freezes the parameters learned from previous tasks.
>
> > Q3
>
> ER mitigates the forgetting of previous tasks by replaying samples stored in the examplar memory, while EWC and PackNet directly enforce preserving parameters of neural networks. Thus, since ER is indirect in transferring a model's knowledge to future tasks, the forward transfer of bias also occurs less for ER than EWC and PackNet, as seen in [Figure 5, Section 5]. However, we note that the lowest BMR for ER observed in [Figure 5, Section 5] does not necessarily imply superior performance in achieving debiased models. This is because the results are not acquired from the optimal hyperparameter in terms of BMR. For a more comprehensive performance comparison, refer to the results presented in [Table 1, Section 6].

---

> > ### Comment · Reviewer_UUYC · 2023-11-22
> > **Response to Rebuttal**
> >
> > I thank the authors for the additional experiments in Appendix F and BMR results with a pretrained model. I recommend adding the pretrained results to the final version of the paper. Further, I do agree that doing a reasonably non-trivial theoretical analyses can be deferred for future work, and encourage the authors to explore this avenue in a follow up. My other concerns and questions are also well addressed.
> >
> > Overall I am convinced that this work presents a meaningful contribution to distribution shifts and continual learning given its fairly rigorous analyses of continual learning on tasks with spurious correlations. I will retain my score (increasing my confidence rating) and recommend paper for acceptance.

---

> > > ### Author Response · Authors · 2023-11-23
> > > **Official comment bu authors**
> > >
> > > Thank you very much for your positive feedback! We promise to add the results for the use of pre-trained models in the final version and further study the bias transfer from a theoretical perspective.

---

### Meta-Review · Area_Chair_6c5x · 2023-12-07

**Metareview:**

(a) Summarize the scientific claims and findings of the paper based on your own reading and characterizations from the reviewers.
- The authors explore the effect of spurious features in continual learning.
- The proposed setting establishes that forward- and backward-transfer are affected if a subset of the tasks in the stream "contain" spurious features
- The authors show that such
- The authors also propose a plug-in method for reducing the negative effect of the spurious features
- The proposed plug-in is found to improve the quality of several standard CL methods across datasets

(b) What are the strengths of the paper?
- The general setting, while synthetic, seems well-motivated
- The empirical analysis provides good motivation for the setting and the method
- The proposed plug-in method shows performance improvements across all tested vision datasets (3) and NLP dataset (1).

(c) What are the weaknesses of the paper? What might be missing in the submission?
- Both the empirical analysis and evaluation are done using (semi-)synthetic data. It is unknown if the proposed setting is realized in more realistic scenarios and whether the observations and findings might carry over.
- [Minor] In (Lesort, 2022) I could not describe precisely the difference between local spurious features and what you refer to as bias. In particular, you write "f local spurious features (LSF) which correlates with class labels in a particular task but vanishes when data samples from all tasks are considered." This seems similar to what is shown in Figure 1. That is, without "unbiased" tasks, models might correctly rely on the features.

**Justification For Why Not Higher Score:**

The paper studies a very synthetic setting, and it is unclear how the findings carry over to more realistic scenarios.

The proposed approach is perhaps more akin to a baseline for the newly proposed scenario. As such, it does not bring significant novelty.

**Justification For Why Not Lower Score:**

I found this paper to be relatively well executed. It might serve (along with two recent references) as a basis for researchers to further explore the applicability of the observed phenomena (effect of spurious features) in more challenging/realistic scenarios.

---

### Decision · Program_Chairs · 2024-01-16

Accept (poster)